# Layerwise Federated Learning for Heterogeneous Quantum Clients using Quorus

**Jason Han, Nicholas S. DiBrita, Daniel Leeds, Jianqiang Li, Jason Ludmir & Tirthak Patel**
Department of Computer Science
Rice University
Houston, TX 77005, USA
`{jh146,nd52,dl107,jl567,jzl2,tp53}@rice.edu`

## Abstract

Quantum machine learning (QML) holds the promise to solve classically intractable problems, but, as critical data can be fragmented across private clients, there is a need for distributed QML in a quantum federated learning (QFL) format. However, the quantum computers that different clients have access to can be error-prone and have heterogeneous error properties, requiring them to run circuits of different depths. We propose a novel solution to this QFL problem, *Quorus*, that utilizes a layerwise loss function for effective training of varying-depth quantum models, which allows clients to choose models for high-fidelity output based on their individual capacity. Quorus also presents various model designs based on client needs that optimize for shot budget, qubit count, midcircuit measurement, and optimization space. Our simulation and real-hardware results show the promise of Quorus: it increases the magnitude of gradients of higher depth clients and improves testing accuracy by 12.4% on average over the state-of-the-art.

## 1 Introduction

*Quantum machine learning (QML)* holds the potential to solve classically difficult problems with high efficiency. Existing methods using quantum ML have been applied to a variety of industrial and scientific applications, including portfolio optimization, drug discovery, and weather forecasting (Peral-García et al., 2024; Smaldone et al., 2025; Liu et al., 2025). Quantum ML has also been used to solve classical ML problems with significant reductions in parameters (Kashif et al., 2025; DiBrita et al., 2025; Leither et al., 2025). Given the success of quantum ML, a natural consideration, like in classical ML, is to consider the real-world case of fragmented data across multiple private clients. *How can clients with quantum computers train together, without revealing data to other parties?* The classical analog of solving this problem has also been proposed, called *Quantum Federated Learning (QFL)* (Chen & Yoo, 2021).

However, existing QFL techniques do not consider the *heterogeneity of quantum devices*. All quantum computers are subject to *hardware error* that varies from computer to computer, which has continued to be a critical challenge in quantum computing (Tannu & Qureshi, 2019; Montanez-Barrera et al., 2025). The quantum computing research community has proposed Quantum Error Correction (QEC) as a solution to quantum hardware error (Calderbank & Shor, 1996; Acharya et al., 2024); however, QEC techniques require millions of qubits, which will not exist for many years (Gidney & Ekerå, 2021; Sevilla & Riedel, 2020). Thus, in our current day, to use error-prone devices for QML tasks, one strategy is to limit the *depth* of the circuit (quantum code) that is executed on the hardware, as the error manifested in the output of the circuit is proportional to its depth. Reducing the depth of the circuit is particularly important, as a major source of quantum errors is *decoherence*, where a qubit loses its important amplitude and phase information with respect to time (Schlosshauer, 2019; Zurek, 2003; Preskill, 2018). *By keeping the circuit to a reasonably shallow depth, researchers attempt to utilize existing quantum computers to achieve practical quantum utility today.*

Another challenge in QML is the *barren plateaus problem*, where gradients vanish as the circuit depth grows (Anschuetz, 2025; Yan et al., 2024; Patel et al., 2024). In the worst case, gradients

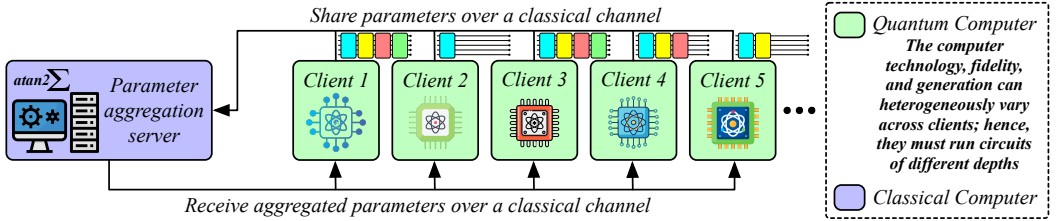

Figure 1: Depiction of the overall setup of our depth-heterogeneous quantum FL framework. Our setup utilizes the realistic scenario of a classical network for sending and receiving parameters, and each client has a quantum computer that can run circuits of varying depths.

decay exponentially, making it practically impossible to train deep circuits, even when noise is not the dominating factor (Cerezo et al., 2021). This significantly restricts the scalability of QML circuits, as optimization becomes infeasible beyond moderate depths. A further obstacle is *resource efficiency*. Unlike classical training, which can rely on inexpensive iterations, every quantum training step requires repeated circuit executions (shots) to estimate observables (McClean et al., 2016). As a result, algorithms must be designed to minimize the number of shots needed for accurate training to ascertain the economic viability for real-world QML.

To address the above challenges, in this work, we design an error-aware QFL technique, *Quorus*, by considering that clients can only run quantum circuits of particular depths, based on the depth at which they can achieve reasonably high accuracy to participate in FL. We illustrate our overall setup in Fig. 1. While existing QFL works have shown that if training is done in the presence of noise that corrupts the output, then the final training accuracy degrades with depth (Rahman et al., 2025; Sahu & Gupta, 2024), our goal is to enable clients to run as many layers as possible, to allow for higher expressive power and thus, higher accuracy (Sim et al., 2019). Quorus is the first-of-its-kind work that utilizes layerwise loss functions and knowledge distillation for synchronized objectives across heterogeneous-depth clients. We propose novel shot-efficient designs for varying quantum hardware capabilities and demonstrate both higher gradient magnitudes as well as implementations on all of IBM's state-of-the-art superconducting quantum computers.

**The contributions of this work are as follows:**

- The first structured quantum federated learning framework (Quorus) that utilizes layerwise losses and reverse distillation for improved accuracy (to the best of our knowledge).
- A quantum model architecture whereby an ensemble of layerwise quantum classifiers can be obtained with no shot overhead, leading to both higher accuracy and resource efficiency.
- A design that improves testing accuracy by up to 12.4% over Q-HeteroFL (Diao et al., 2021)while being shot-efficient for different binary classification tasks.
- A model that yields higher gradient norms, reducing barren plateau effect, and achieves accuracy within 3% of ideal simulation on IBM superconducting QPUs, showing real-world viability.

## 2 PRELIMINARIES

**Quantum Computing Basics.** Quantum computers process information by manipulating qubits with quantum circuits. The *state* of a qubit is represented as a vector: $|\psi\rangle = \beta_0 |0\rangle + \beta_1 |1\rangle$, where $\beta_0, \beta_1 \in \mathbb{C}$ and $|\beta_0|^2 + |\beta_1|^2 = 1$. The state $|\psi\rangle$ exists in a superposition of the states $|0\rangle$ and $|1\rangle$, which encodes the quantum data we process. The probability of measuring the qubit to be in state $|i\rangle$ is $p(i) = |\beta_i|^2$, meaning we must have $|\beta_0|^2 + |\beta_1|^2 = 1$. A *statevector* of a system of $n$ qubits is a complex vector in the Hilbert space $|\psi\rangle \in \mathbb{C}^{2^n} = \mathcal{H}_n$, that is normalized $\langle\psi|\psi\rangle = 1$. We can write our state in the computational basis: defining $b_k$ as the bitstring corresponding to the integer $k$, the computational basis is the set $\{|b_k\rangle \, \forall k \in \mathbb{Z}, 0 \le k \le 2^{n-1}\}$. Our state can be expressed as $|\psi\rangle = \sum_{k=0}^{2^{n-1}} \beta_k |k\rangle$. The quantum data $|\psi\rangle$ is processed by a quantum circuit $U$, a unitary operator taking $U |\psi_1\rangle = |\psi_2\rangle$. Because $U$ is unitary, it is reversible ($UU^\dagger = U^\dagger U = I$).

**Parameterized Quantum Circuits.** To frame a learning problem on quantum computers, we parameterize the operation $U$ with parameters $\theta$, which are typically rotation angles on the Bloch sphere. Then, $U(\theta)$ is a *parameterized quantum circuit* (PQC) with trainable gate parameters, and

the structure of $U(\theta)$ is referred to as an *ansatz*. These variational quantum circuits are often composed of repeated circuit structures called *layers*, and can be written as $U(\theta) = U_{0:L}(\theta_{0:L}) = U_L(\theta_L)U_{L-1}(\theta_{L-1})...U_0(\theta_0)$, where $U_i$ is the parameterized circuit for layer $i$, with parameters $\theta_i$. Deeper circuits are more expressive, but also suffer from decoherence and errors when evaluated on real hardware. Quantum machine learning generally aims to solve the following problem for an objective $L$ and input data $x$: $\theta^\star = \arg\min_{\theta \in \Theta} L(U, x; \theta)$. To evaluate $L(U, x; \theta)$ on a quantum computer, it is performed by estimating $p(b)$, the probability of measuring state $|b\rangle$ via running the circuit $U(\theta)$ multiple times, and tracking how many times the outcome $b$ was observed. Each run of the quantum circuit is called a *shot*, and shots are expensive on current-day quantum hardware.

**Quantum Measurements.** The objective $L$ of a parameterized circuit is extracted via projective measurement, which is irreversible in general. For a particular outcome $b \in \{0, 1\}$, if the first qubit in state $|\psi\rangle$ is measured to be $b$, then the resulting state is collapsed to $|\psi_b\rangle = \frac{1}{\sqrt{p(b)}} \sum_{k:k_1=b} \beta_k |k\rangle$.

This fundamental quantum property poses a unique challenge when the objective $L_j$ is defined for each layer, so $L_j = L(U_{0:j}; \theta_{0:j})$, where $U_{0:j}, \theta_{0:j}$ represents the layers and parameters up to layer $j$. Because measuring a qubit collapses the superposition and removes information from the quantum state, it poses a challenge for simultaneously collapsing information via measurement and retaining sufficient information for subsequent quantum layers and operations (Gyawali et al., 2024).

**Heterogeneous Federated Learning.** *Federated learning* (FL) is a distributed machine learning technique widely used in classical ML where each client's data is private to themselves (McMahan et al., 2023). The overall objective function in federated learning for $m$ clients is $L(x_1, x_2, ..., x_m) = \frac{1}{m} \sum_{i=1}^{m} L_i(x_i)$, where $x_j$ represents the data of client $j$ and $L_j$ is the loss function for client $j$ (McMahan et al., 2023). In *centralized federated learning*, training is done locally by clients, and parameters are aggregated in a centralized server and broadcast back to clients. An intuition may be gained for why parameter aggregation works by observing that, in the special case where stochastic gradient descent (SGD) is used, parameters are aggregated every epoch, and the batch size is equal to the amount of data a client has, it is equivalent in expectation to performing SGD on the centralized objective $L$ (McMahan et al., 2023).

*Heterogeneous Federated Learning* adds a layer of complexity to FL by allowing for clients to have different local model architectures (Diao et al., 2021). This scenario accounts for the case where some clients have differential computational abilities, but still want to take advantage of FL to obtain a shared model. Because the parameter spaces of models are now different, special considerations need to be made as the differing model architectures lead to *parameter mismatches* that can negatively affect training (Kim et al., 2023). Refer to Appendix A for further details.

## 3   RELATED WORK

**Classical Federated Learning.** The problem of depth-heterogeneous quantum federated learning, where clients have classical models, has a large body of work in the literature, but many state-of-the-art techniques in classical FL cannot be directly applied to the case where the model is a PQC. The classical model-heterogeneous FL technique, HeteroFL (Diao et al., 2021), aggregates parameters in shared submodels across clients. Since this original work, some newer techniques have been proposed, namely FEDepth and ScaleFL (Zhang et al., 2025; Ilhan et al., 2023), which assume that intermediate layers can be trained; however, this is not applicable to PQC's as training these layers requires a client to run circuits to that depth, which precisely is the bottleneck in quantum circuits.

Another work, ReeFL, uses a transformer to fuse features between layers; however, features are not directly accessible in quantum ML unless via state tomography (Lee et al., 2024). The classical work most closely related to Quorus, DepthFL (Kim et al., 2023), is amenable to the setup of quantum clients with models of varying depths as it is a layerwise FL technique; however, evaluating the layerwise loss function on quantum computers is nontrivial due to measurement collapse, which we discuss further in Sec. 4.3. Overall, these classical works *cannot* be directly applied to the quantum FL setup and highlight the importance of quantum-centric design, which we propose in this work.

**Quantum Federated Learning.** The overall setup of QFL, where clients use the same architecture PQC and use a centralized server for parameter aggregation, has been studied (Chen & Yoo, 2021); however, the problem of depth-heterogeneous FL is not well studied in the quantum case. One work that tackles the problem of parameters being lost in communication, named eSQFL (Yun et al.,

2022), uses a layerwise loss function by computing the inner product of states between each layer; however, computing inner products requires long-distance connectivity and is not applicable to run on real hardware (Sá et al., 2023). *There is a gap in the literature related to quantum federated learning for clients with heterogeneous needs, which we propose a solution for in this work.*

## 4    DESIGN AND APPROACH

The overall workflow of our technique is illustrated in Fig. 1. Each client is able to train a PQC of a different depth based on its hardware capability. After local training, clients send their parameters to a server over a classical network, where parameters are aggregated and sent back to clients. Clients then continue to train locally, repeating the process for a set number of rounds.

---

**Algorithm 1:** Quorus

---

**Initialization :** $\theta^0$
**Server Executes:**
$P \leftarrow$ All Clients
**for** *round* $t = 0, 1, \ldots, T - 1$ **do**
    $\theta^{t+1} \leftarrow 0$
    **forall** $k \in P$ *(in parallel)* **do**
        $\tilde{\theta}^t \leftarrow \theta^t[: d_k]$
        $\tilde{\theta}_k^{t+1} \leftarrow$ **Client_Update**$(k, \tilde{\theta}^t)$
        $\theta^{t+1}[: d_k] \leftarrow \theta^{t+1}[: d_k] + e^{i\tilde{\theta}_k^{t+1}}$
    **foreach** *resource capability* $d_i$ **do**
        $\theta^{t+1}[d_i] \leftarrow$ **angle**$\left(\frac{1}{\left|P^{d_k \geq d_i}\right|} \theta^{t+1}[d_i]\right)$

**Client_Update**$(k, \tilde{\theta}^t)$**:**
$\tilde{\theta}_k^{t+1} \leftarrow \tilde{\theta}^t$
**for** *local epoch* $e = 1, 2, \ldots, E$ **do**
    **for** *each mini-batch* $b_h$ **do**
        $L_k = \sum_{i=1}^{d_k} L_{ce}^i + \frac{1}{d_k - 1} \sum_{i=1}^{d_k} \sum_{\substack{j=1 \\ j \neq i}}^{d_k} \mathrm{D_{KL}}(p_j \,\|\, p_i)$
        $\tilde{\theta}_k^{t+1} \leftarrow \tilde{\theta}_k^{t+1} -$ **Adam**$(\nabla L_k(\tilde{\theta}_k^{t+1}; b_h), \eta, h)$
**return** $\tilde{\theta}_k^{t+1}$

---

### 4.1    QUORUS WORKFLOW

A detailed description of our workflow is depicted in Algorithm 1 (Kim et al., 2023). Because PQC's have heterogeneous depths, parameters are aggregated only among the clients that share each parameter. We also perform aggregation of parameters with circular averaging because the quantum circuit parameters in our implementation are rotation angles, where the *angle* function is defined as $angle(z) = atan2(imag(z), real(z))$. The layerwise loss function for client $k$ is

$$L_k = \sum_{i=1}^{d_k} L_{ce}^i + \frac{1}{d_k - 1} \sum_{i=1}^{d_k} \sum_{j=1,\ j\neq i}^{d_k} \mathrm{D_{KL}}(p_j \,\|\, p_i), \tag{1}$$

where $L_{ce}^i$ is the Binary Cross Entropy loss for the classifier at depth $i$. Note, then, that this loss function assumes that there is a means to extract classifier outputs at each layer – an important problem with a unique quantum design (addressed in Sec. 4.3). $d_k$ is the depth of client $k$ and $\mathrm{D_{KL}}(p_j \,\|\, p_i)$ is the KL divergence between logits $p_i$ and $p_j$. We use the same loss function as in DepthFL (Kim et al., 2023), because a similar intuition applies: we want a loss function that clients share to address *parameter mismatching*, where parameters are different across clients due to varying local parameter spaces. In addition, we want to use the KL divergence for "reverse distillation", whereby deeper classifiers are helped by shallower ones. These nuances are explored and justified in  (Kim et al., 2023), so we do not repeat the discussion for the quantum case.

The unique challenge in the quantum case is how exactly to evaluate the loss in Eq. 1. In our setup for Quorus, the loss $L_{ce}^i$ is computed via the probability of measuring a qubit to be 0 or 1 for our binary classification tasks. This poses unique quantum-specific design considerations for Quorus, which the following sections will be devoted to solving.

### 4.2    ANSATZ DESIGNS AND SELECTION

For any quantum computing problem, it is well-known that deciding the ansatz is essential (Sugisaki et al., 2022), for two main reasons. Firstly, it determines the expressibility and the space of solution states that are explored, and because of the exponentially-sized Hilbert space that quantum computers operate in, operating in a relevant subspace is essential (Sim et al., 2019; Yan et al., 2024; Holmes et al., 2022). Secondly, it is entirely possible to find an ansatz that is well-suited for the problem of interest, but is very inefficient when implemented on hardware architectures with limited connectivity due to its use of long-distance two-qubit or multi-qubit gates, causing high levels

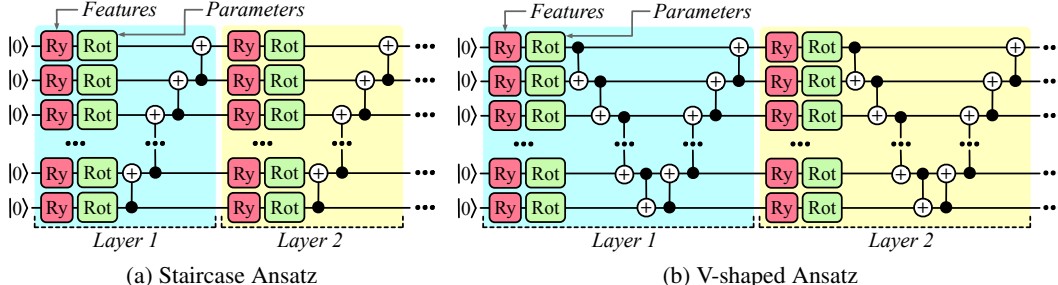

(a) Staircase Ansatz          (b) V-shaped Ansatz

Figure 2: We evaluate three ansätze for Quorus: (1) The staircase ansatz, (2) the V-shaped ansatz, and (3) the alternating ansatz, which switches between staircase and V-shaped layers (not shown).

of output error (Kivlichan et al., 2018; Romero et al., 2018). We address these problems by systematically exploring relevant ansätze for our problem, depicted in Fig. 2. In each layer of our ansatz, we perform *data reuploading* as it has been shown in multiple quantum ML experiments to yield improved accuracy and nonlinearities with respect to the input (Vidal & Theis, 2020; Aminpour et al., 2024). We use the *Ry* gate to achieve this. We use layers of generalized single-qubit gates *Rot* as tunable parameters. The ansatzes we design are centered around two main principles:

(1) The ansatz must be hardware efficient, so we assume only nearest-neighbor connectivity as observed in quantum hardware Huo et al. (2025); Han et al. (2025), and (2) only the first qubit is measured to obtain the output statistic. The latter choice is because we focus on binary classification tasks in this work, so measuring a single qubit is sufficient (Schuld et al., 2020). Thus, we come up with two relevant ansätze: the first is the *Staircase* ansatz, depicted in Fig. 2(a) (Schuld et al., 2020; Sim et al., 2019), which has a staircase

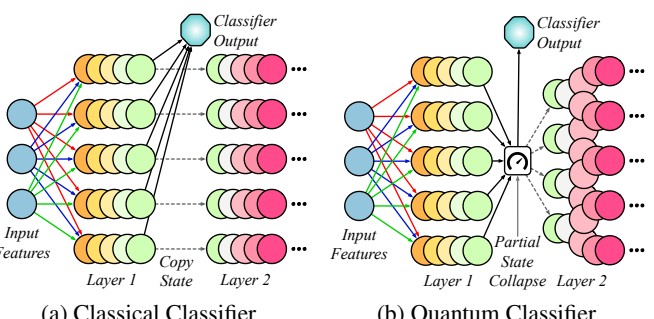

(a) Classical Classifier     (b) Quantum Classifier

Figure 3: The difference between classical and quantum layerwise classifiers. Measurements collapse quantum data, and thus an altered state is passed to the next layer.

of CNOTs from the last qubit up to the first one. The second is the *V-shaped* ansatz, which has a staircase of CNOTs going down from the first to the last qubit, which then go back up to the first qubit, and the third is an alternating combination of the two. Based on our experimental evaluation on various datasets in (Appendix D), we observe that the V-shaped ansatz performs the best in a majority of the evaluations, and so we use it as the default ansatz. The reason that the V-shaped ansatz performs well is its ability to broadcast information throughout qubits with more CNOT gates traversing up and down the qubits (Sim et al., 2019).

## 4.3 QUANTUM CLASSIFIER DESIGN

When one attempts to implement the layerwise loss function in Eq. 1, there is an immediate problem: if we measure the qubit, then how can we pass on the same state to the next layer? An illustration of this dilemma is in Fig. 3. In DepthFL and other classical works that assume an intermediate classifier, depicted in Fig. 3(a), the data after the first layer is somehow converted to a scalar, and implicitly, the data is passed, unchanged, to the next layer (and this "copy" operation has minimal classical overhead). Fundamentally, a direct analog does not exist in quantum computation. In quantum computing, if we measure one of the qubits mid-computation, this collapses the

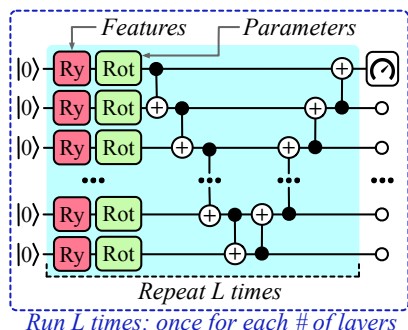

Figure 4: The Quorus-Layerwise design. The circuit must be run $L$ times, where $L$ is the number of layers.

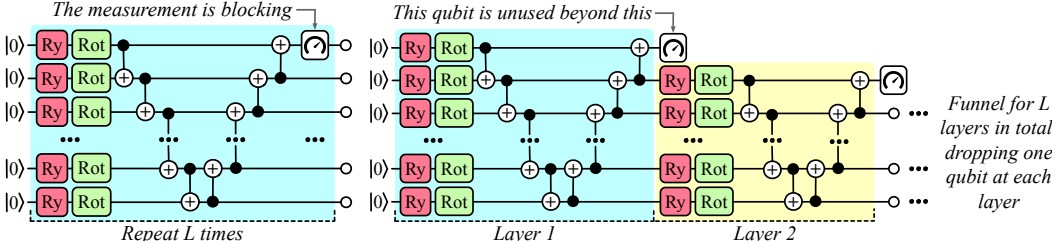

(a) Quorus-Blocking Circuit            (b) Quorus-Funnel Circuit

Figure 6: (a) The Blocking design (logically $\equiv$ to the Ancilla design), and (b) the Funnel design. Blocking requires a midcircuit measurement, and Funnel restricts the size of unitary operations.

superposition on the first state and changes the state that is later used in computation (as represented in Fig. 3(b)). Passing the state unchanged thus requires you to prepare another copy of it, which induces additional shot overhead that is linear in the number of layers and is a nontrivial cost, given the expense of running quantum computers. For example, running quantum circuits for just one minute on an IBM quantum computer costs $96 (can run $\approx 4$ circuits in this time with 1k shots each) ibm. Thus, we are posed with an important question: *How do we implement this measurement between layers in a quantum ML model, in a shot-efficient manner?*

**Solution 1: Layerwise.** The most straightforward solution to the classical case is to "copy" the quantum state, because we know exactly the circuit that prepared it. This solution is depicted in Fig. 4. However, this requires a shot budget that scales linearly with the number of layers. For deep circuits, the required shot budget quickly becomes infeasible for budget-constrained clients.

**Solution 2: Ancilla/Blocking.** To address the case where a client does not have a high-shot budget, we design an ansatz where it is possible to obtain predictions from each layer with shots independent of the layer count. In particular, with reference to Figure 3(b), we propose continuing to operate on the collapsed state in our PQC. This design is depicted in Fig. 5. In particular, we entangle the first qubit with an ancilla in the $|0\rangle$ state after each layer. We

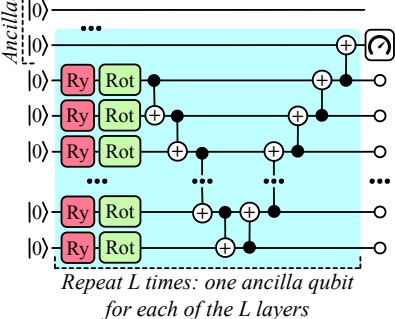

Figure 5: The Quorus-Ancilla design. The circuit is only run once, but requires an ancilla qubit per layer, and also dephases the first qubit.

evaluate each layer's outputs by computing the marginal distribution on its ancilla. For the first layer, the statistics match the Layerwise model; for later layers, they differ because entangling the first qubit with an ancilla "dephases" it (Gyawali et al., 2024).

Since dephasing is limited to that qubit, we hypothesize, and confirm on IBM hardware (Sec. 5), that our quantum ML model can still train effectively under this alternative model. Implementing this requires the first qubit to entangle with a new ancilla at each layer, which in turn demands long-distance CNOTs. Thus, while our Layerwise ansatz assumes nearest-neighbor connectivity, systems with larger qubit counts and richer connectivity can benefit from this Ancilla ap-

Table 1: Unique requirements of different quantum models of Quorus, highlighting their usecases.

| Model | ↑ Shot Budget | ↑ Qubit Count | Midcirc. Meas. | ↓ Hilbert Space |
|---|---|---|---|---|
| Layerwise | ✓ | ✗ | ✗ | ✗ |
| Ancilla | ✗ | ✓ | ✗ | ✗ |
| Blocking | ✗ | ✗ | ✓ | ✗ |
| Funnel | ✗ | ✗ | ✗ | ✓ |

proach. In principle, ancillae are not required – one can simply just measure the first qubit, not reset it, and continue in computation. This logically equivalent (proof provided in Appendix B), but physically distinct model of computation is depicted in Fig. 6(a), where a midcircuit measurement is performed on the first qubit. This model would be feasible for clients who can do fast midcircuit measurements, but existing midcircuit measurements are lengthy and error-prone (Deist et al., 2022; Rudinger et al., 2021).

**Solution 3: Funnel.** Finally, for clients that may not have a high-shot budget, no ancillas, and no midcircuit measurement capability, we design a model that layer-by-layer drops operations that act

Table 2: Capacity-wise Comparison (V-Shape) — Baselines + Quorus-Layerwise with $\Delta$ to the Best (**bolded**). The means and standard deviations are shown for five different samples of data. We see that Quorus-Layerwise consistently outperforms the baselines across client capacities.

| Capacity | Technique | MNIST | | | Fashion-MNIST | | |
|---|---|---|---|---|---|---|---|
| | | 0/1 | 3/4 | 4/9 | Trouser/Boot | Bag/Sandal | Pullover/Coat |
| 2L | Q-HeteroFL | 90.3 ± 5.2 (↓ 7.9) | 58.8 ± 12.6 (↓ 37.3) | 59.3 ± 7.3 (↓ 20.7) | 62.4 ± 33.5 (↓ 36.4) | 67.1 ± 14.6 (↓ 24.8) | 58.9 ± 6.1 (↓ 18.0) |
| | Vanilla QFL (2L) | 98.2 ± 0.4 (↓ 0.0) | 96.0 ± 1.2 (↓ 0.1) | **80.0 ± 0.9** | 98.5 ± 1.1 (↓ 0.3) | 91.9 ± 1.2 | **76.9 ± 0.4** |
| | Standalone | **98.2 ± 0.3** | **96.1 ± 1.2** | 78.5 ± 2.5 (↓ 1.5) | 98.3 ± 0.9 (↓ 0.5) | 91.2 ± 1.0 (↓ 0.7) | 74.9 ± 1.8 (↓ 2.0) |
| | Quorus-Layerwise | 97.0 ± 1.4 (↓ 1.2) | 95.0 ± 1.2 (↓ 1.1) | 78.2 ± 0.6 (↓ 1.8) | **98.8 ± 0.9** | 86.1 ± 8.1 (↓ 5.8) | 76.3 ± 1.4 (↓ 0.6) |
| 3L | Q-HeteroFL | 79.6 ± 14.8 (↓ 18.7) | 85.0 ± 3.8 (↓ 11.9) | 68.5 ± 4.7 (↓ 11.9) | 76.9 ± 15.0 (↓ 22.3) | 79.1 ± 12.0 (↓ 13.0) | 59.5 ± 10.5 (↓ 19.1) |
| | Vanilla QFL (2L) | 98.2 ± 0.4 (↓ 0.1) | 96.0 ± 1.2 (↓ 0.9) | 80.0 ± 0.9 (↓ 0.4) | 98.5 ± 1.1 (↓ 0.7) | 91.9 ± 1.2 (↓ 0.2) | 76.9 ± 0.4 (↓ 1.7) |
| | Standalone | **98.3 ± 1.2** | 95.5 ± 1.7 (↓ 1.4) | 79.6 ± 3.4 (↓ 0.8) | 99.1 ± 0.6 (↓ 0.1) | **92.1 ± 2.4** | 76.5 ± 2.5 (↓ 2.1) |
| | Quorus-Layerwise | 98.0 ± 1.0 (↓ 0.3) | **96.9 ± 0.7** | **80.4 ± 2.4** | **99.2 ± 0.4** | 89.2 ± 5.9 (↓ 2.9) | **78.6 ± 1.0** |
| 4L | Q-HeteroFL | 80.4 ± 7.6 (↓ 17.9) | 88.0 ± 7.3 (↓ 9.5) | 68.8 ± 5.0 (↓ 13.1) | 90.5 ± 6.6 (↓ 8.8) | 88.6 ± 2.0 (↓ 5.1) | 72.8 ± 3.2 (↓ 5.9) |
| | Vanilla QFL (2L) | 98.2 ± 0.4 (↓ 0.1) | 96.0 ± 1.2 (↓ 1.5) | 80.0 ± 0.9 (↓ 1.9) | 98.5 ± 1.1 (↓ 0.8) | 91.9 ± 1.2 (↓ 1.8) | 76.9 ± 0.4 (↓ 1.8) |
| | Standalone | 98.0 ± 2.5 (↓ 0.3) | 97.4 ± 0.5 (↓ 0.1) | 81.2 ± 3.7 (↓ 0.7) | 98.9 ± 1.1 (↓ 0.4) | **93.7 ± 1.1** | 77.1 ± 1.0 (↓ 1.6) |
| | Quorus-Layerwise | **98.3 ± 0.9** | **97.5 ± 0.6** | **81.9 ± 2.2** | **99.3 ± 0.3** | 91.5 ± 4.0 (↓ 2.2) | **78.7 ± 1.0** |
| 5L | Q-HeteroFL | 89.9 ± 3.7 (↓ 8.6) | 88.5 ± 5.3 (↓ 9.0) | 71.0 ± 4.4 (↓ 11.5) | 94.6 ± 4.3 (↓ 4.7) | 88.4 ± 3.8 (↓ 5.9) | 72.8 ± 1.2 (↓ 6.0) |
| | Vanilla QFL (2L) | 98.2 ± 0.4 (↓ 0.3) | 96.0 ± 1.2 (↓ 1.5) | 80.0 ± 0.9 (↓ 2.5) | 98.5 ± 1.1 (↓ 0.8) | 91.9 ± 1.2 (↓ 2.4) | 76.9 ± 0.4 (↓ 1.9) |
| | Standalone | 97.2 ± 1.7 (↓ 1.3) | 96.4 ± 1.9 (↓ 1.1) | 80.3 ± 3.2 (↓ 2.2) | 98.5 ± 0.5 (↓ 0.8) | **94.3 ± 0.6** | 77.6 ± 1.6 (↓ 1.2) |
| | Quorus-Layerwise | **98.5 ± 0.8** | **97.5 ± 0.4** | **82.5 ± 2.5** | **99.3 ± 0.2** | 92.4 ± 2.6 (↓ 1.9) | **78.8 ± 1.1** |
| 6L | Q-HeteroFL | 88.6 ± 6.2 (↓ 10.0) | 85.1 ± 5.9 (↓ 12.7) | 73.9 ± 4.4 (↓ 9.2) | 95.3 ± 0.8 (↓ 4.1) | 92.1 ± 1.3 (↓ 3.2) | 74.4 ± 0.9 (↓ 4.4) |
| | Vanilla QFL (2L) | 98.2 ± 0.4 (↓ 0.4) | 96.0 ± 1.2 (↓ 1.8) | 80.0 ± 0.9 (↓ 3.1) | 98.5 ± 1.1 (↓ 0.9) | 91.9 ± 1.2 (↓ 3.4) | 76.9 ± 0.4 (↓ 1.9) |
| | Standalone | 98.3 ± 1.0 (↓ 0.3) | 96.5 ± 0.8 (↓ 1.3) | 80.4 ± 3.4 (↓ 2.7) | 98.3 ± 0.8 (↓ 1.1) | **95.3 ± 1.0** | 75.4 ± 1.5 (↓ 3.4) |
| | Quorus-Layerwise | **98.6 ± 0.8** | **97.8 ± 0.2** | **83.1 ± 2.4** | **99.4 ± 0.3** | 92.7 ± 2.5 (↓ 2.6) | **78.8 ± 0.8** |

on the first qubit, allowing all measurements to be at the end (Killoran et al., 2019). This model is depicted in Fig. 6(b), where we gradually "funnel" down the size of the deeper unitaries by dropping a qubit after each measurement, hence the name of this technique. The cost of this model is that the user must have a problem that is amenable to operating on fewer and fewer qubits.

**Ansatz Use Case.** We summarize the costs associated with each ansatz design in Table 1 to highlight their unique usecases. Note that each model has disjoint requirements – that is, each model has exactly one cost, thus highlighting the versatility of our design choices to clients' unique scenarios. We evaluate each design in Sec. 5 to compare their accuracy performance.

## 5 EXPERIMENTAL EVALUATION

Here, we evaluate Quorus on different binary classification tasks. A comprehensive description of the experimental setup is in Appendix C. We evaluate with 128 datapoints per client, consistent with existing QFL literature; to account for the random sampling of the data, we evaluate each of our comparisons with five different runs with five different samples of data allocations for clients. Data is reduced to 10 features using Principal Component Analysis (PCA), the 10-dimensional data is angle-encoded with RY gates, and the data is reuploaded using RY gates for each layer.

**Quorus outperforms state-of-the-art techniques in terms of classification accuracy.** The state-of-the-art baselines that compare Quorus against are informed by what setups clients could run given that each has a different depth model, based on existing techniques described in Sec. 3. We compare against: (1) *Q-HeteroFL*, our quantum version of a classical technique called HeteroFL (Diao et al., 2021). Here, all clients run the maximum depth model they can, and the parameters are averaged only over the clients that contain those parameters. This work does not explicitly consider heterogeneous-model federated learning using PQC. Thus, our design of the quantum version of HeteroFL itself is novel and described in Appendix C. (2) *Vanilla QFL*, where all clients use the same depth model as the shallowest-depth client. For clients that are able to run deeper models, they are unable to fully utilize their quantum resources. (3) *Standalone*, where clients do not participate in the FL process and train the data on their own. This approach has the clear disadvantage that clients do not get the benefit of training an improved model from other clients' data.

We present our results in Table 2, comparing the baselines above to Quorus-Layerwise, because it uses the same model architecture as the baselines. We present our results from the perspective of the client of different capacities in the leftmost "Capacity" column – for that capacity, what is the best performing technique for the various class comparisons? We see that, across client capacities, for most comparisons, Quorus-Layerwise has the highest mean testing accuracy (12.4% over Q-Hetero-FL). Notably, for clients of the smallest capacity, Quorus-Layerwise does not have the highest testing accuracy, but this is likely due to its modified loss function, which penalizes the first layer parameters, along with the loss values for clients with later layers. Another important observation is that

Table 3: Capacity-wise Comparison (V-Shape) — Quorus Variants Only with $\Delta$ to the Best (**bolded**). The means and standard deviations are calculated over five runs: Quorus-Layerwise and Quorus-Funnel have the highest testing accuracy (we use them for subsequent experiments).

| Capacity | Technique | MNIST | | | Fashion-MNIST | | |
|---|---|---|---|---|---|---|---|
| | | 0/1 | 3/4 | 4/9 | Trouser/Boot | Bag/Sandal | Pullover/Coat |
| 2L | Quorus-Layerwise | 97.0 ± 1.4 (↓ 0.4) | 95.0 ± 1.2 (↓ 0.3) | 78.2 ± 0.6 (↓ 1.5) | **98.8 ± 0.9** | 86.1 ± 8.1 (↓ 0.3) | 76.3 ± 1.4 (↓ 0.2) |
| | Quorus-Ancilla/Blocking | 97.0 ± 1.0 (↓ 0.4) | 94.8 ± 1.5 (↓ 0.5) | 78.3 ± 1.2 (↓ 1.4) | 98.6 ± 0.9 (↓ 0.2) | 86.2 ± 8.4 (↓ 0.2) | **76.5 ± 1.3** |
| | Quorus-Funnel | **97.4 ± 1.2** | **95.3 ± 1.5** | **79.7 ± 2.0** | 98.1 ± 1.1 (↓ 0.7) | **86.4 ± 6.8** | 76.4 ± 1.2 (↓ 0.1) |
| 3L | Quorus-Layerwise | 98.0 ± 1.0 (↓ 0.1) | 96.9 ± 0.7 (↓ 0.0) | 80.4 ± 2.4 (↓ 1.9) | **99.2 ± 0.4** | 89.2 ± 5.9 (↓ 1.0) | 78.6 ± 1.0 (↓ 0.1) |
| | Quorus-Ancilla/Blocking | 97.9 ± 1.2 (↓ 0.2) | **96.9 ± 0.6** | 81.4 ± 2.0 (↓ 0.9) | 99.2 ± 0.5 (↓ 0.0) | 88.7 ± 7.5 (↓ 1.5) | 78.5 ± 1.2 (↓ 0.2) |
| | Quorus-Funnel | **98.1 ± 0.5** | 96.9 ± 0.6 (↓ 0.0) | **82.3 ± 1.8** | 98.9 ± 0.6 (↓ 0.3) | **90.2 ± 3.3** | **78.7 ± 1.1** |
| 4L | Quorus-Layerwise | 98.3 ± 0.9 (↓ 0.0) | **97.5 ± 0.6** | 81.9 ± 2.2 (↓ 1.3) | **99.3 ± 0.3** | 91.5 ± 4.0 (↓ 0.8) | 78.7 ± 1.0 (↓ 0.7) |
| | Quorus-Ancilla/Blocking | 98.1 ± 1.2 (↓ 0.2) | 97.3 ± 0.5 (↓ 0.2) | 81.5 ± 1.9 (↓ 1.7) | 99.2 ± 0.5 (↓ 0.1) | 90.3 ± 5.5 (↓ 2.0) | 78.9 ± 1.0 (↓ 0.5) |
| | Quorus-Funnel | **98.3 ± 0.7** | 97.1 ± 0.6 (↓ 0.4) | **83.2 ± 2.4** | 99.0 ± 0.6 (↓ 0.3) | **92.3 ± 1.7** | **79.4 ± 1.0** |
| 5L | Quorus-Layerwise | 98.5 ± 0.8 (↓ 0.0) | **97.5 ± 0.4** | 82.5 ± 2.5 (↓ 2.1) | **99.3 ± 0.2** | 92.4 ± 2.6 (↓ 0.3) | 78.8 ± 1.1 (↓ 1.5) |
| | Quorus-Ancilla/Blocking | 98.3 ± 0.7 (↓ 0.2) | 97.4 ± 0.5 (↓ 0.1) | 81.9 ± 2.5 (↓ 2.7) | 99.3 ± 0.3 (↓ 0.0) | 91.1 ± 4.3 (↓ 1.6) | 79.0 ± 1.4 (↓ 1.4) |
| | Quorus-Funnel | **98.5 ± 0.7** | 97.1 ± 0.5 (↓ 0.4) | **84.6 ± 2.1** | 99.1 ± 0.4 (↓ 0.2) | **92.7 ± 1.2** | **80.4 ± 1.0** |
| 6L | Quorus-Layerwise | **98.6 ± 0.8** | **97.8 ± 0.3** | 83.1 ± 2.4 (↓ 2.1) | **99.4 ± 0.3** | 92.7 ± 2.5 (↓ 0.7) | 78.8 ± 0.8 (↓ 1.5) |
| | Quorus-Ancilla/Blocking | 98.4 ± 0.6 (↓ 0.2) | 97.5 ± 0.5 (↓ 0.3) | 82.2 ± 2.0 (↓ 3.0) | 99.3 ± 0.3 (↓ 0.1) | 91.4 ± 4.0 (↓ 2.0) | 78.8 ± 1.1 (↓ 1.5) |
| | Quorus-Funnel | 98.0 ± 0.7 (↓ 0.6) | 97.1 ± 0.4 (↓ 0.7) | **85.2 ± 0.7** | 99.1 ± 0.4 (↓ 0.3) | **93.4 ± 0.9** | 80.3 ± 0.9 |

Figure 7: We show the per-layer magnitude of the gradients for Quorus-Layerwise by plotting the mean and standard deviation of the gradient norms for each epoch (smoothed for readability). Compared to Q-HeteroFL, we see that our modified loss function has larger gradient norms throughout training for parameters earlier in the circuit, due to earlier measurements.

Q-HeteroFL performs substantially worse compared to Quorus-Layerwise, sometimes nearly 40% worse, as in MNIST 3/4 classification. This is due to the *parameter mismatching* challenge: different, conflicting loss functions lead to suboptimal models. *This highlights the importance of having a shared loss function between clients that can be optimized (Quorus-Layerwise).*

**The performance of the different variants of Quorus.** We now evaluate the performance of the variants of Quorus in Table 3. Note that, because the Quorus-Ancilla and Quorus-Blocking designs are logically equivalent, their accuracies are displayed together. We see that Quorus-Layerwise and Quorus-Funnel have the best testing accuracy, although for many class comparisons, the difference between the techniques is within a single percentage point. This suggests that all of the Quorus have high testing accuracy, and that the decision of which model to use depends on the resource constraints of the client, as mentioned in Table 1. In particular, for the Quorus-Funnel model, we observe that, even though later unitaries operate in a smaller Hilbert space, the testing accuracy is always within 1% of the best performing Quorus design, indicating its comparable accuracy. Importantly, the Quorus-Funnel model is also shot-efficient, and for the hardest classes (MNIST 4/9, Fashion-MNIST Pullover/Coat), it performs the best compared to the other models. Thus, we use the Quorus-Funnel model for subsequent noise analysis on real hardware evaluations.

## 6 ANALYSIS OF QUORUS'S FUNCTIONALITY

We split our analysis into two types: (1) gradient norms analysis and (2) analysis on real IBM superconducting hardware. Due to the extensive amount of runs required and the prohibitive cost of real-hardware runs, we only provide this analysis for the Pullover/Coat classification task from the Fashion-MNIST dataset as this is the most challenging task.

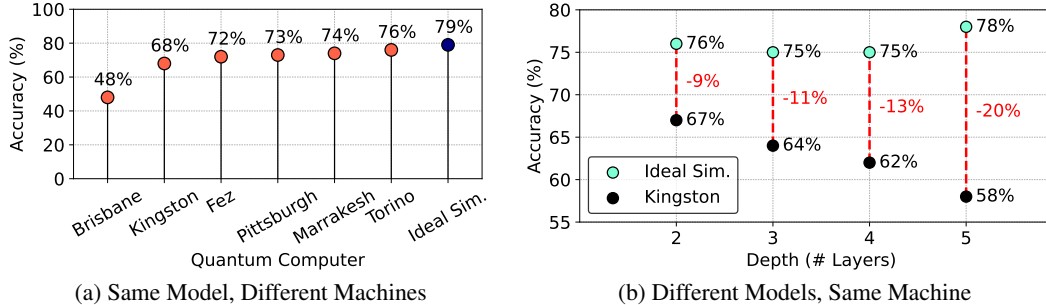

(a) Same Model, Different Machines  (b) Different Models, Same Machine

Figure 8: With the Quorus-Funnel model, we show (a) how the same model has varying testing accuracy based on the real machine used, and (b) how using smaller depths leads to higher testing accuracy on a machine. This highlights the importance of clients using depths that they can accurately evaluate the model on, as well as the practical hardware relevance of our experimental setup.

Table 4: Capacity-wise Comparison (V-Shape) — Q-HeteroFL vs Quorus variants on CIFAR-10 with $\Delta$ to the Best (**Bolded**). The means and standard deviations are calculated over five runs: Quorus-Layerwise and Quorus-Funnel outperform Q-HeteroFL on all comparisons.

| Capacity | Technique | CIFAR-10 | | |
|---|---|---|---|---|
| | | Deer/Truck | Automobile/Truck | Cat/Dog |
| 2L | Q-HeteroFL | $65.0 \pm 5.8$ ($\downarrow 11.7$) | $55.8 \pm 3.1$ ($\downarrow 5.9$) | $50.5 \pm 1.0$ ($\downarrow 4.9$) |
| | Quorus-Layerwise | $\mathbf{76.7 \pm 1.8}$ | $61.0 \pm 2.9$ ($\downarrow 0.7$) | $\mathbf{55.4 \pm 1.5}$ |
| | Quorus-Funnel | $76.5 \pm 1.9$ ($\downarrow 0.2$) | $\mathbf{61.7 \pm 1.9}$ | $54.7 \pm 0.8$ ($\downarrow 0.7$) |
| 3L | Q-HeteroFL | $63.7 \pm 8.2$ ($\downarrow 13.5$) | $53.8 \pm 2.8$ ($\downarrow 8.3$) | $52.8 \pm 1.3$ ($\downarrow 2.9$) |
| | Quorus-Layerwise | $76.9 \pm 1.6$ ($\downarrow 0.3$) | $61.7 \pm 1.5$ ($\downarrow 0.4$) | $\mathbf{55.7 \pm 1.5}$ |
| | Quorus-Funnel | $\mathbf{77.2 \pm 1.7}$ | $\mathbf{62.1 \pm 1.3}$ | $55.6 \pm 1.0$ ($\downarrow 0.1$) |
| 4L | Q-HeteroFL | $72.1 \pm 2.6$ ($\downarrow 6.2$) | $56.5 \pm 4.2$ ($\downarrow 6.0$) | $51.6 \pm 2.2$ ($\downarrow 3.9$) |
| | Quorus-Layerwise | $77.5 \pm 1.3$ ($\downarrow 0.8$) | $62.4 \pm 1.0$ ($\downarrow 0.1$) | $\mathbf{55.5 \pm 1.4}$ |
| | Quorus-Funnel | $\mathbf{78.3 \pm 1.4}$ | $\mathbf{62.5 \pm 1.5}$ | $55.1 \pm 0.8$ ($\downarrow 0.4$) |
| 5L | Q-HeteroFL | $68.0 \pm 5.1$ ($\downarrow 10.4$) | $55.8 \pm 1.2$ ($\downarrow 6.9$) | $52.8 \pm 1.7$ ($\downarrow 2.9$) |
| | Quorus-Layerwise | $77.7 \pm 1.3$ ($\downarrow 0.7$) | $62.6 \pm 1.0$ ($\downarrow 0.1$) | $\mathbf{55.7 \pm 1.2}$ |
| | Quorus-Funnel | $\mathbf{78.4 \pm 1.7}$ | $\mathbf{62.7 \pm 1.4}$ | $55.4 \pm 1.6$ ($\downarrow 0.3$) |
| 6L | Q-HeteroFL | $70.6 \pm 2.6$ ($\downarrow 7.8$) | $57.3 \pm 1.6$ ($\downarrow 5.3$) | $51.5 \pm 1.4$ ($\downarrow 4.1$) |
| | Quorus-Layerwise | $77.9 \pm 1.3$ ($\downarrow 0.6$) | $\mathbf{62.6 \pm 1.0}$ | $\mathbf{55.6 \pm 1.1}$ |
| | Quorus-Funnel | $\mathbf{78.4 \pm 2.1}$ | $62.3 \pm 1.5$ ($\downarrow 0.3$) | $55.2 \pm 1.1$ ($\downarrow 0.4$) |

**Higher Gradient Norms with Quorus.** Typically, the deeper the circuit, the smaller the magnitudes of the gradients become (Cerezo et al., 2021). We verify this empirically in our setup as well, plotting the gradient norms of Quorus-Layerwise and Q-HeteroFL in Fig. 7. We see that for the Q-HeteroFL model, the gradient norms are small for each layer in the quantum circuit, as the loss for all parameters is defined on the output measurement at the end of the circuit. Interestingly, for Quorus, this is not the case. Because we have loss functions that are defined after each layer, for deep models, earlier layers maintain a high gradient norm due to these layerwise loss functions. Note that for parameters in later layers, the gradient norms remain small because the gradients for these parameters only depend on measurements deep in the circuit. However, the overall magnitudes of the gradients for Quorus are higher, and in addition, Quorus also obtains a higher testing accuracy than Q-HeteroFL, making it implausible that the reasons for the larger gradient norms are due to a lack of convergence for Quorus. This result suggests how the layerwise loss function in Quorus can be used for improved and scalable trainability for deep quantum circuits.

**Evaluation of Quorus on Real Quantum Hardware.** We evaluate our trained models on all of IBM's superconducting quantum processing units or QPUs to demonstrate the practical relevance of our experimental setup, as well as the very real impact that noise has on our trained models. In particular, we perform our hardware analysis using the Quorus-Funnel design, because it allows for evaluating the ensembled model with no shot overhead. Due to the high error of midcircuit measurements on current hardware, we measure all qubits at the end (Rudinger et al., 2021; Gao

Table 5: Capacity-wise Comparison (V-Shape) — Q-HeteroFL vs Quorus variants on 4-class MNIST/Fashion-MNIST with Δ to the Best (**Bolded**). The means and standard deviations are calculated over five runs: Quorus-Layerwise and Quorus-Funnel outperform Q-HeteroFL on all tasks.

| Capacity | Technique | MNIST | | | Fashion-MNIST | | |
|---|---|---|---|---|---|---|---|
| | | 0/1/6/7 | 0/2/4/7 | 3/5/8/9 | Trouser/Sandal/Sneaker/Bag | Pullover/Dress/Coat/Boot | Top/Pullover/Coat/Shirt |
| 2L | Q-HeteroFL | $30.7 \pm 15.0$ ($\downarrow$ 46.0) | $33.9 \pm 17.5$ ($\downarrow$ 31.3) | $35.5 \pm 9.6$ ($\downarrow$ 20.4) | $32.2 \pm 8.2$ ($\downarrow$ 46.4) | $43.4 \pm 12.8$ ($\downarrow$ 27.0) | $36.4 \pm 6.0$ ($\downarrow$ 12.5) |
| | Quorus-Layerwise | $73.2 \pm 5.2$ ($\downarrow$ 3.5) | $64.6 \pm 7.4$ ($\downarrow$ 0.6) | **$55.9 \pm 3.8$** | $65.5 \pm 9.3$ ($\downarrow$ 13.1) | $65.6 \pm 4.2$ ($\downarrow$ 4.8) | $44.7 \pm 4.6$ ($\downarrow$ 4.2) |
| | Quorus-Funnel | **$76.7 \pm 3.7$** | **$65.2 \pm 10.1$** | $55.0 \pm 5.9$ ($\downarrow$ 0.9) | **$78.6 \pm 3.5$** | **$70.4 \pm 1.6$** | **$48.9 \pm 3.4$** |
| 3L | Q-HeteroFL | $39.5 \pm 12.9$ ($\downarrow$ 41.4) | $46.3 \pm 8.7$ ($\downarrow$ 32.9) | $35.4 \pm 5.0$ ($\downarrow$ 25.6) | $50.3 \pm 10.4$ ($\downarrow$ 30.5) | $38.8 \pm 12.6$ ($\downarrow$ 35.3) | $36.0 \pm 5.2$ ($\downarrow$ 20.1) |
| | Quorus-Layerwise | **$80.9 \pm 2.7$** | $71.1 \pm 6.2$ ($\downarrow$ 8.1) | **$61.0 \pm 4.7$** | $71.4 \pm 6.4$ ($\downarrow$ 9.4) | $69.1 \pm 3.8$ ($\downarrow$ 5.0) | $52.2 \pm 2.6$ ($\downarrow$ 3.9) |
| | Quorus-Funnel | $80.1 \pm 4.6$ ($\downarrow$ 0.8) | **$79.2 \pm 6.7$** | $60.6 \pm 5.4$ ($\downarrow$ 0.4) | **$80.8 \pm 4.8$** | **$74.1 \pm 1.2$** | **$56.1 \pm 3.0$** |
| 4L | Q-HeteroFL | $53.9 \pm 13.9$ ($\downarrow$ 29.4) | $50.6 \pm 9.4$ ($\downarrow$ 27.7) | $48.1 \pm 4.7$ ($\downarrow$ 17.6) | $56.4 \pm 10.4$ ($\downarrow$ 25.5) | $55.1 \pm 6.9$ ($\downarrow$ 19.0) | $40.1 \pm 5.7$ ($\downarrow$ 17.0) |
| | Quorus-Layerwise | **$83.3 \pm 4.2$** | $76.5 \pm 5.3$ ($\downarrow$ 1.8) | **$65.7 \pm 2.4$** | $75.5 \pm 5.8$ ($\downarrow$ 6.4) | $72.1 \pm 3.8$ ($\downarrow$ 2.0) | $54.1 \pm 0.8$ ($\downarrow$ 3.0) |
| | Quorus-Funnel | $81.1 \pm 4.0$ ($\downarrow$ 2.2) | **$78.3 \pm 7.2$** | $63.6 \pm 4.4$ ($\downarrow$ 2.1) | **$81.9 \pm 4.5$** | **$74.1 \pm 2.8$** | **$57.1 \pm 2.4$** |
| 5L | Q-HeteroFL | $58.0 \pm 20.7$ ($\downarrow$ 27.0) | $54.4 \pm 13.8$ ($\downarrow$ 24.2) | $42.8 \pm 3.5$ ($\downarrow$ 24.6) | $54.6 \pm 11.0$ ($\downarrow$ 27.7) | $47.3 \pm 5.5$ ($\downarrow$ 27.6) | $49.1 \pm 4.1$ ($\downarrow$ 9.0) |
| | Quorus-Layerwise | **$85.0 \pm 3.3$** | $78.0 \pm 3.8$ ($\downarrow$ 0.6) | $66.2 \pm 2.7$ ($\downarrow$ 1.2) | $78.5 \pm 4.6$ ($\downarrow$ 3.8) | $73.5 \pm 2.9$ ($\downarrow$ 1.4) | $55.4 \pm 0.8$ ($\downarrow$ 2.7) |
| | Quorus-Funnel | $80.4 \pm 3.9$ ($\downarrow$ 4.6) | **$78.6 \pm 6.8$** | **$67.4 \pm 3.1$** | **$82.3 \pm 3.3$** | **$74.9 \pm 3.2$** | **$58.1 \pm 2.9$** |
| 6L | Q-HeteroFL | $71.7 \pm 6.3$ ($\downarrow$ 14.5) | $71.4 \pm 2.8$ ($\downarrow$ 8.1) | $53.3 \pm 2.5$ ($\downarrow$ 14.3) | $71.3 \pm 2.5$ ($\downarrow$ 11.2) | $60.6 \pm 7.5$ ($\downarrow$ 14.6) | $46.9 \pm 3.7$ ($\downarrow$ 10.5) |
| | Quorus-Layerwise | **$86.2 \pm 2.9$** | **$79.5 \pm 3.3$** | **$67.6 \pm 2.9$** | $80.1 \pm 4.2$ ($\downarrow$ 2.4) | $74.3 \pm 2.2$ ($\downarrow$ 0.9) | $56.0 \pm 1.1$ ($\downarrow$ 1.4) |
| | Quorus-Funnel | $78.3 \pm 5.2$ ($\downarrow$ 7.9) | $77.0 \pm 7.7$ ($\downarrow$ 2.5) | $66.6 \pm 3.3$ ($\downarrow$ 1.0) | **$82.5 \pm 2.4$** | **$75.2 \pm 3.8$** | **$57.4 \pm 3.2$** |

et al., 2025). The depth of the Quorus-Funnel models therefore matters, as for deeper models, more decoherence will accumulate on the qubits that are unused or carry information from earlier layers.

**(A) Same Model, Different QPUs.** We evaluate Quorus with a depth of 5 on different IBM QPUs to validate the heterogeneity of quantum systems. We restricted our testing set to only 100 datapoints due to the prohibitive cost of each shot. Our results in Fig. 8(a) show that across six different QPUs, the noise varies substantially: from 48% for IBM Brisbane to 76% for IBM Torino, 3% off from the ideal simulation accuracy. If a client has access to a machine with similar hardware noise characteristics to IBM Torino, they should go with a deep circuit. Thus, we observe a diverse spectrum of error on IBM's QPUs, validating the practical relevance of our experimental setup.

**(B) Different Depth Model, Same QPU.** In Fig. 8(a), we notice that the QPU with the lowest testing accuracy (aside from IBM Brisbane, which suffered from decoherence to 48% testing accuracy with just two layers of our model) is IBM Kingston. Thus, we performed an analysis on IBM Kingston to verify the impact of decreasing the depth of the quantum circuit on the testing accuracy; we expect that as we reduce the number of layers, the testing accuracy should increase. Our hypothesis is empirically validated in Fig. 8(b). We see that the separation, indicated by a dashed red line, between the ideal simulation results and the testing accuracy on IBM Kingston gets wider with a deeper circuit. This highlights that, for a client with access to a computer similar to IBM Kingston in terms of hardware noise, it is advantageous for them to train a shallow-depth quantum classifier, because these classifiers have lower-error outputs and can more meaningfully contribute to FL.

## 7 ABLATION: ADDITIONAL CLASSIFICATION TASKS

To evaluate Quorus on more complex classification tasks, we additionally run Quorus-Layerwise and Quorus-Funnel for binary classification on CIFAR-10 in Table 4 and four-class classification for MNIST/Fashion-MNIST data in Table 5 (categorical cross-entropy loss is used; further details are in Appendix I.2). We see an average improvement of 6.7% in CIFAR-10 binary classification and 24.0% in MNIST/Fashion-MNIST four-class classification of the best performing variant of Quorus over Q-HeteroFL, highlighting Quorus's robustness to more challenging classification tasks.

## 8 CONCLUSION

In this work, we introduced Quorus, a QFL framework tailored for heterogeneous-depth clients. Our contributions include: (1) a layerwise loss with high gradient norms to align objectives across clients of varying circuit depths, (2) multiple circuit designs, Layerwise, Ancilla/Blocking, and Funnel, that balance accuracy with resource constraints, and (3) extensive evaluation showing up to 12.4% accuracy improvements over Q-HeteroFL and consistently higher gradient magnitudes for deeper clients. Crucially, we validated Quorus on all of IBM's superconducting quantum processors, demonstrating that our method is not only effective in simulation but also practical on today's error-prone hardware. *Together, these results establish Quorus as the first implementable framework for QFL in realistic multi-client settings, paving the way for scalable and resource-aware QML.*

ACKNOWLEDGEMENT

This work was supported by Rice University, the Rice University George R. Brown School of Engineering and Computing, and the Rice University Department of Computer Science. This work was supported by the DOE Quantum Testbed Finder Award DE-SC0024301, the Ken Kennedy Institute, and Rice Quantum Initiative, which is part of the Smalley-Curl Institute. We acknowledge the use of IBM Quantum services for this work. The views expressed are those of the authors and do not reflect the official policy or position of IBM or the IBM Quantum team.

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

## A    FURTHER PRELIMINARIES

**Quantum Gates.**  Quantum gates are unitary operators that manipulate qubits in a quantum circuit, analogous to logic gates in classical circuits. The *rotation gates* apply continuous rotations of a qubit's state on the Bloch sphere (Fig. 9). For example, the $R_y(\theta)$ gate performs a rotation around the $y$-axis by angle $\theta$:

$$R_y(\theta) \;=\; \begin{bmatrix} \cos\left(\frac{\theta}{2}\right) & -\sin\left(\frac{\theta}{2}\right) \\ \sin\left(\frac{\theta}{2}\right) & \cos\left(\frac{\theta}{2}\right) \end{bmatrix}.$$

More generally, the three-axis rotation operator $Rot(\alpha, \beta, \gamma)$ applies successive rotations about the $x$, $y$, and $z$ axes by angles $\alpha$, $\beta$, and $\gamma$:

$$Rot(\alpha, \beta, \gamma) \;=\; R_z(\alpha)\, R_y(\beta)\, R_z(\gamma),$$

where

$$R_z(\phi) = \begin{bmatrix} e^{-i\phi/2} & 0 \\ 0 & e^{i\phi/2} \end{bmatrix}.$$

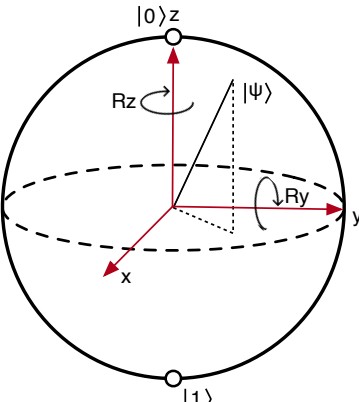

Figure 9: Visualization of rotation gates on a quantum Bloch sphere.

Entangling gates act on two or more qubits. A key example is the controlled-NOT (CNOT) gate, which flips the target qubit if the control qubit is in state $|1\rangle$. Its $4 \times 4$ unitary matrix is:

$$\mathrm{CNOT} \;=\; \begin{bmatrix} 1 & 0 & 0 & 0 \\ 0 & 1 & 0 & 0 \\ 0 & 0 & 0 & 1 \\ 0 & 0 & 1 & 0 \end{bmatrix}.$$

The gate representation in a circuit diagram has the target qubit and the control qubit connected with a vertical line, with the control qubit indicated by a filled-in circle and the target qubit indicated by the $\oplus$ symbol. Together, single-qubit rotation gates and entangling gates like CNOT form a universal gate set, capable of approximating any quantum operation.

## B    PROOF THAT THE QUORUS-ANCILLA CIRCUIT IS EQUIVALENT TO THE QUORUS-BLOCKING CIRCUIT

In this section, we prove that the circuits in Fig. 5 and Fig. 6(a), namely, the Ancilla technique and the blocking technique, are equivalent. To do so, we will consider the state of both circuits immediately after the measurement operation.

We first note that both circuits apply the same unitary $U |0\rangle^{\otimes n} = |\psi\rangle$. We note that the same logic applies for subsequent layers (replacing $|\psi\rangle$ with the resulting input state will suffice), so analyzing this single-layer setup is sufficient.

Additionally, we assume that, in the Ancilla circuit, the ancilla qubit is immediately measured after the CNOT gate. This simplifies the analysis and is equivalent to the case where the ancilla is measured later, as no other operations are performed on the ancilla qubit, so we apply the deferred measurement principle (Gurevich & Blass, 2021).

**Proposition 1** (Ancilla–measurement equals measuring the control). *Let $U$ be an $n$-qubit unitary and let $|\psi\rangle = U |0\rangle^{\otimes n}$. Write $|\psi\rangle$ as*

$$|\psi\rangle = \alpha \, |0\rangle \, |\phi_0\rangle + \beta \, |1\rangle \, |\phi_1\rangle \,,$$

*where the first ket is qubit 0, $|\phi_b\rangle$ are normalized states of the remaining $n - 1$ qubits, and $|\alpha|^2 + |\beta|^2 = 1$. Consider two procedures:*

Table 6: Experimental details, specification, and hyperparameters used to evaluate Quorus.

| Parameter | Value |
|---|---|
| Dataset | MNIST; Fashion-MNIST |
| Classes | 0/1, 3/4, 4/9 for MNIST; Trouser/Boot, Bag/Sandal, Pullover/Coat for Fashion-MNIST |
| Number of clients $K$ | 5 |
| Datapoints per Client | 128 |
| Testing set size | 3000; 100 for hardware runs only |
| Data Distribution | IID |
| Number of different data splits per class comparison | 5 |
| Data encoding scheme | Angle Embedding |
| Client sampling per round | 100% |
| Communication rounds $T$ | 1000 |
| Local epochs per round $E$ | 1 |
| Batch size | 32 |
| Optimizer | Adam ($\beta_1$=0.9, $\beta_2$=0.99) |
| Learning rate $\eta$ | 0.001 |
| LR schedule | 1.0 (No decay) |
| Loss type | Binary Cross Entropy on Labels KL Divergence between logits |
| Aggregation Method | Circular averaging of subnet parameters |
| Qubits $q$ | 10 |
| Depth levels $L$ | 2L, 3L, 4L, 5L, 6L |
| Parameters per layer | 30 |
| Parameter Initialization | $\mathcal{N}(0,1)$ |

(A) Direct measurement. *Measure qubit $0$ in the computational ($Z$) basis. The probabilities are $p_b = \|\left(|b\rangle\langle b| \otimes I\right)|\psi\rangle\|^2 = |\alpha|^2$ for $b = 0$ and $|\beta|^2$ for $b = 1$, and the post-measurement (normalized) states of the remaining qubits are $|\phi_b\rangle$.*

(B) Ancilla measurement. *Prepare an ancilla $a$ in $|0\rangle_a$, apply a CNOT with control qubit $1$ and target $a$, then measure $a$ in the computational $Z$ basis. After the CNOT, the joint state is*

$$\alpha\,|0\rangle\,|0\rangle_a\,|\phi_0\rangle + \beta\,|1\rangle\,|1\rangle_a\,|\phi_1\rangle.$$

*Projecting onto $|b\rangle_a$ yields outcome $b$ with probability $p'_b = \|\alpha\,|0\rangle\,|\phi_0\rangle\|^2$ for $b = 0$ and $\|\beta\,|1\rangle\,|\phi_1\rangle\|^2$ for $b = 1$, i.e. $p'_0 = |\alpha|^2$ and $p'_1 = |\beta|^2$. Conditioned on outcome $b$, the (normalized) post-measurement state of the system qubits is $|b\rangle\,|\phi_b\rangle$; tracing out qubit $1$ leaves the remaining qubits in $|\phi_b\rangle$.*

*Thus, $p_b = p'_b$ and the conditional post-measurement states of the non-ancilla qubits coincide in (A) and (B). Consequently, for any subsequent (classically controlled) processing, the two procedures are operationally equivalent.* □

## C DETAILS OF OUR EXPERIMENTAL METHODOLOGY

We present the experimental details and hyperparameters in Table 6. Because image data is high-dimensional and amplitude encoding is not feasible on near-term devices due to the high depth (Han et al., 2025), we perform angle encoding using RY gates. For each layer of the quantum circuit, data is reuploaded with angle encoding using RY gates. This means that we must compress the image into a set of 10 features, and we do so using PCA. One might ask the question of how to perform PCA on decentralized data. This problem has been solved using a technique called Federated PCA (Grammenos et al., 2020), which provides the same PCA results as centralized PCA. Because the implementation details of Federated PCA are not central to Quorus, we emulate

Table 7: IBM QPUs and their performance characteristics. 2Q refers to the two-qubit gate error, which is typically specified, as it is an order of magnitude more dominant than the 1Q gate error.

| QPU name | Qubits | 2Q error (best) | 2Q error (layered) | CLOPS | Processor type |
|---|---|---|---|---|---|
| Pittsburgh | 156 | 8.11E-4 | 3.81E-3 | 250K | Heron r3 |
| Kingston | 156 | 7.82E-4 | 3.57E-3 | 250K | Heron r2 |
| Fez | 156 | 1.45E-3 | 4.28E-3 | 195K | Heron r2 |
| Marrakesh | 156 | 1.11E-3 | 3.72E-3 | 195K | Heron r2 |
| Torino | 133 | 1.29E-3 | 7.50E-3 | 210K | Heron r1 |
| Brisbane | 127 | 2.87E-3 | 1.74E-2 | 180K | Eagle r3 |

Federated PCA with centralized PCA in our implementation and note that the PCA implementation can be substituted as desired.

For inference on testing data, the testing data is compressed using the PCA fit on the training data. We assume the use of Federated PCA for all of our experiments, even for Standalone training, for both consistency and for considering the "adversarial" case where a client decides to participate in Federated PCA to obtain better reduced features, but chooses not to participate in the FL process. In addition, to consider the most adversarial setup for Standalone training, where a client does not participate in the FL process, the optimizer state for Adam persists across rounds (whereas, in our QFL setups, we reset the Adam optimizer state each round, as done in Wang et al. (2021)). We evaluate on the specific classes in MNIST and Fashion-MNIST as they represent various levels of difficulty, used in other QML works (DiBrita et al., 2025; Ranjan et al., 2024).

The hardware specifications of the IBM QPUs are provided in Table 7.

### C.1 Q-HETEROFL FRAMEWORK

---

**Algorithm 2:** Q-HeteroFL

---

**Initialization :** $\theta^0$

**Server Executes:**
$P \leftarrow$ All Clients
**for** *round* $t = 0, 1, \ldots, T - 1$ **do**
    $\theta^{t+1} \leftarrow 0$
    **forall** $k \in P$ *(in parallel)* **do**
        $\tilde{\theta}^t \leftarrow \theta^t[: d_k]$
        $\tilde{\theta}_k^{t+1} \leftarrow$ **Client_Update**$(k, \tilde{\theta}^t)$
        $\theta^{t+1}[: d_k] \leftarrow \theta^{t+1}[: d_k] + e^{i\tilde{\theta}_k^{t+1}}$
    **foreach** *resource capability* $d_i$ **do**
        $\theta^{t+1}[d_i] \leftarrow$ **angle**$\left(\frac{1}{\left|P^{d_k \geq d_i}\right|} \theta^{t+1}[d_i]\right)$

**Client_Update**$(k, \tilde{\theta}^t)$**:**
$\tilde{\theta}_k^{t+1} \leftarrow \tilde{\theta}^t$
**for** *local epoch* $e = 1, 2, \ldots, E$ **do**
    **for** *each mini-batch* $b_h$ **do**
        $L_k = L_{\text{ce}}^{d_k}$
        $\tilde{\theta}_k^{t+1} \leftarrow \tilde{\theta}_k^{t+1} - \mathbf{Adam}(\nabla L_k(\tilde{\theta}_k^{t+1}; b_h), \eta, h)$
**return** $\tilde{\theta}_k^{t+1}$

---

We describe the Q-HeteroFL technique in Algorithm 2, an adaptation of the aggregation technique for heterogeneous classical FL described in Diao et al. (2021). The loss function is defined solely on the deepest classifier output, and aggregation is also done using circular averaging for consistency in comparison to Quorus. HeteroFL is the standard baseline in heterogeneous FL but has not yet been proposed in QFL; thus, we propose it here and demonstrate Quorus's improvements over it.

## D  ADDITIONAL RESULTS AND ANALYSIS

### D.1  ANSATZ CHOICE ANALYSIS

We perform a comprehensive analysis of what ansatz to use in our experiments for Quorus-Layerwise by evaluating the Staircase, V-shape, and an Alternating variant of the former two across MNIST and Fashion-MNIST classes. Note that, for $L$ layers in our Quorus-Layerwise, we have $L - 1$ different classifiers (one classifier per layer, with the first layer having two variational layers). This means that, for a client that can run a capacity of $L$ layers, they can ensemble the outputs of their $L - 1$ classifiers for inference. That is what is shown in Table 8 and is how Quorus is evaluated

Table 8: Best Ansatz by Client Capacity for Ensembled Submodels, *Quorus*. The table is sectioned off into different capacities based on the number of layers a client can run. The best-performing ansatz for each different class comparison is in **bold**. The V-shaped ansatz has the highest testing accuracy the most times, so we use it for all of our experiments.

| Capacity | Ansatz | MNIST | | | Fashion-MNIST | | |
|---|---|---|---|---|---|---|---|
| | | 0/1 | 3/4 | 4/9 | Trouser/Boot | Bag/Sandal | Pullover/Coat |
| 2L | Staircase | **97.9 ± 1.1** | **95.2 ± 1.9** | 73.5 ± 5.3 (↓ 4.7) | 97.5 ± 2.1 (↓ 1.3) | **93.4 ± 0.9** | 66.9 ± 1.7 (↓ 9.4) |
| | V-shape | 97.0 ± 1.4 (↓ 0.9) | 95.0 ± 1.2 (↓ 0.2) | **78.2 ± 0.6** | **98.8 ± 0.9** | 86.1 ± 8.1 (↓ 7.3) | **76.3 ± 1.4** |
| | Alternating | 88.2 ± 6.6 (↓ 9.7) | 84.5 ± 17.6 (↓ 10.7) | 65.5 ± 5.8 (↓ 12.7) | 82.0 ± 19.8 (↓ 16.8) | 82.7 ± 16.3 (↓ 10.7) | 66.8 ± 4.9 (↓ 9.5) |
| 3L | Staircase | **98.2 ± 1.0** | 96.2 ± 0.8 (↓ 0.7) | **81.0 ± 2.8** | 98.6 ± 0.8 (↓ 0.6) | **93.7 ± 0.8** | 74.1 ± 3.0 (↓ 4.5) |
| | V-shape | 98.0 ± 1.0 (↓ 0.2) | **96.9 ± 0.7** | 80.4 ± 2.4 (↓ 0.6) | **99.2 ± 0.4** | 89.2 ± 5.9 (↓ 4.5) | **78.6 ± 1.0** |
| | Alternating | 92.0 ± 4.0 (↓ 6.2) | 94.8 ± 1.2 (↓ 2.1) | 79.9 ± 1.6 (↓ 1.1) | 97.1 ± 0.7 (↓ 2.1) | 91.7 ± 0.9 (↓ 2.0) | 73.5 ± 3.4 (↓ 5.1) |
| 4L | Staircase | 98.2 ± 0.6 (↓ 0.1) | 96.4 ± 0.7 (↓ 1.1) | **82.0 ± 2.3** | 98.7 ± 0.9 (↓ 0.6) | **93.7 ± 0.9** | 73.8 ± 2.4 (↓ 4.9) |
| | V-shape | **98.3 ± 0.9** | **97.5 ± 0.6** | 81.9 ± 2.2 (↓ 0.1) | **99.3 ± 0.3** | 91.5 ± 4.0 (↓ 2.2) | **78.7 ± 1.0** |
| | Alternating | 93.8 ± 2.0 (↓ 4.5) | 95.2 ± 1.1 (↓ 2.3) | 81.8 ± 3.0 (↓ 0.2) | 97.6 ± 0.5 (↓ 1.7) | 92.5 ± 1.3 (↓ 1.8) | 74.8 ± 2.3 (↓ 3.9) |
| 5L | Staircase | 98.1 ± 0.6 (↓ 0.4) | 96.3 ± 0.4 (↓ 1.2) | **83.0 ± 2.6** | 98.8 ± 0.7 (↓ 0.5) | **93.7 ± 0.8** | 74.2 ± 2.2 (↓ 4.6) |
| | V-shape | **98.5 ± 0.8** | **97.5 ± 0.4** | 82.5 ± 2.5 (↓ 0.5) | **99.3 ± 0.2** | 92.4 ± 2.6 (↓ 1.3) | **78.8 ± 1.1** |
| | Alternating | 93.8 ± 2.0 (↓ 4.7) | 95.6 ± 1.1 (↓ 1.9) | 82.1 ± 2.9 (↓ 0.9) | 97.6 ± 0.7 (↓ 1.7) | 92.5 ± 1.2 (↓ 1.2) | 75.3 ± 2.0 (↓ 3.5) |
| 6L | Staircase | 98.0 ± 0.8 (↓ 0.6) | 96.3 ± 0.7 (↓ 1.5) | 83.1 ± 3.2 (↓ 0.0) | 98.8 ± 0.8 (↓ 0.6) | **93.8 ± 0.9** | 74.6 ± 1.9 (↓ 4.2) |
| | V-shape | **98.6 ± 0.8** | **97.8 ± 0.2** | **83.1 ± 2.4** | **99.4 ± 0.3** | 92.7 ± 2.5 (↓ 1.1) | **78.8 ± 0.8** |
| | Alternating | 93.1 ± 2.7 (↓ 5.5) | 95.3 ± 1.1 (↓ 2.5) | 82.4 ± 2.7 (↓ 0.7) | 97.4 ± 0.7 (↓ 2.0) | 92.6 ± 1.1 (↓ 1.2) | 75.3 ± 1.9 (↓ 3.5) |

in the tables in the main text. We see that, across a majority of the capacities and class comparisons, the V-shape ansatz has the highest testing accuracy, making it the better choice on average. A reason for this is that the V-shape has the largest number of CNOT gates and circuit depth compared to the Staircase and Alternating ansatzes, and thus it may be more expressive. From the results in this table, we decide to use the V-shape ansatz as the default in our experiments.

## D.2 ABLATION ON THE NUMBER OF LAYERS

To justify the layer count we used in our experiments, we evaluate Quorus-Layerwise using both fewer and more layers, depicted in Table 9. We run two additional ablations: Quorus with the five clients having 1, 2, 3, 4, and 5 layers respectively; and Quorus with the five clients having 2, 4, 6, 8, and 10 layers, respectively (note that the case where the 5 clients have 2, 3, 4, 5, and 6 layers, respectively is what is used by default in our work).

We would like to point out that using 1 layer appears to have drastically lower testing accuracy, at times 40% lower than 6 or 10 layers. This suggests that clients with 1 layer do not have enough parameters to contribute well to

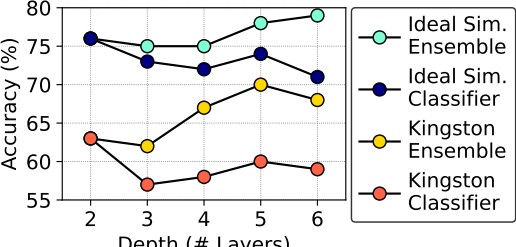

Figure 10: Testing accuracies of the subclassifiers and submodels of a single Quorus - Funnel model evaluated on IBM Kingston. We see that ensembling outputs yields higher accuracy, similar to what we see in ideal simulation.

the training, a result consistent with intuition. There is also a question of whether we use more layers and whether it is helpful for clients. We see that, for our setup, using more layers (up to 8 or 10 layers) has marginal gains in testing accuracy. This result is consistent with quantum computing literature, where adding more layers to solve a problem saturates in gains beyond a certain point Nguyen et al. (2022). Thus, we use 2 through 6 layers in our experimental setup, as more layers lead to higher testing accuracy in this regime, as well as for the fact that quantum circuits of this size are amenable to running on real-world hardware, as we show in our analysis section.

## D.3 ROBUSTNESS OF QUORUS AMIDST REAL HARDWARE NOISE

In comparing the performance of various depth circuits used in Quorus-Funnel on real hardware, we observe an interesting result. In Fig. 10, we plot the testing accuracy on 100 datapoints for one model trained on Fashion-MNIST **Pullover/Coat** classification, evaluated on IBM Kingston. We run our depth 5 model on IBM Kingston, meaning that in total, we extract 5 classifier outputs (one output

Table 9: Capacity-wise Comparison (V-Shape) — Quorus-Layerwise sizes with $\Delta$ to the Best. Means $\pm$ standard deviation shown over five splits; mean ties broken on standard deviation.

| Capacity | Quorus Layer Count | MNIST | | | Fashion-MNIST | | |
|---|---|---|---|---|---|---|---|
| | | 0/1 | 3/4 | 4/9 | Trouser/Boot | Bag/Sandal | Pullover/Coat |
| 1 | Quorus-Layerwise (1L) | $59.5 \pm 1.2$ ($\downarrow 37.5$) | $63.8 \pm 2.7$ ($\downarrow 31.2$) | $67.1 \pm 5.0$ ($\downarrow 11.9$) | $58.6 \pm 2.9$ ($\downarrow 40.2$) | $59.7 \pm 4.5$ ($\downarrow 28.9$) | $66.9 \pm 1.2$ ($\downarrow 9.4$) |
| | Quorus-Layerwise (2L) | $97.0 \pm 1.4$ | $95.0 \pm 1.2$ ($\downarrow 0.0$) | $78.2 \pm 0.6$ ($\downarrow 0.8$) | $98.8 \pm 0.9$ | $86.1 \pm 8.1$ ($\downarrow 2.5$) | $76.3 \pm 1.4$ |
| | Quorus-Layerwise (2L) | $95.0 \pm 3.8$ ($\downarrow 2.0$) | $95.0 \pm 2.3$ | $79.0 \pm 2.7$ | $97.1 \pm 2.3$ ($\downarrow 1.7$) | $88.6 \pm 1.7$ | $75.3 \pm 1.7$ ($\downarrow 1.0$) |
| 2 | Quorus-Layerwise (2L) | $96.4 \pm 1.9$ ($\downarrow 1.6$) | $95.1 \pm 1.9$ ($\downarrow 1.8$) | $77.7 \pm 5.0$ ($\downarrow 4.9$) | $97.3 \pm 1.5$ ($\downarrow 1.9$) | $88.0 \pm 4.7$ ($\downarrow 4.1$) | $75.3 \pm 2.0$ ($\downarrow 3.3$) |
| | Quorus-Layerwise (3L) | $98.0 \pm 1.0$ | $96.9 \pm 0.7$ | $80.4 \pm 2.4$ ($\downarrow 2.2$) | $99.2 \pm 0.4$ | $89.2 \pm 5.9$ ($\downarrow 2.9$) | $78.6 \pm 1.0$ |
| | Quorus-Layerwise (4L) | $97.5 \pm 1.3$ ($\downarrow 0.5$) | $96.7 \pm 1.0$ ($\downarrow 0.2$) | $82.6 \pm 2.1$ | $98.6 \pm 0.7$ ($\downarrow 0.6$) | $92.1 \pm 0.7$ | $77.6 \pm 1.3$ ($\downarrow 1.0$) |
| 3 | Quorus-Layerwise (3L) | $97.5 \pm 1.4$ ($\downarrow 0.8$) | $96.6 \pm 1.1$ ($\downarrow 0.9$) | $80.0 \pm 4.0$ ($\downarrow 3.0$) | $97.9 \pm 1.0$ ($\downarrow 1.4$) | $91.9 \pm 2.3$ ($\downarrow 1.2$) | $77.1 \pm 1.6$ ($\downarrow 1.6$) |
| | Quorus-Layerwise (4L) | $98.3 \pm 0.9$ | $97.5 \pm 0.6$ | $81.9 \pm 2.2$ ($\downarrow 1.1$) | $99.3 \pm 0.3$ | $91.5 \pm 4.0$ ($\downarrow 1.6$) | $78.7 \pm 1.0$ |
| | Quorus-Layerwise (6L) | $97.5 \pm 1.1$ ($\downarrow 0.8$) | $97.0 \pm 0.6$ ($\downarrow 0.5$) | $83.0 \pm 2.3$ | $98.8 \pm 1.0$ ($\downarrow 0.5$) | $93.1 \pm 0.5$ | $78.3 \pm 1.6$ ($\downarrow 0.4$) |
| 4 | Quorus-Layerwise (4L) | $98.5 \pm 0.7$ ($\downarrow 0.0$) | $97.0 \pm 1.0$ ($\downarrow 0.5$) | $81.4 \pm 3.8$ ($\downarrow 2.3$) | $98.9 \pm 0.5$ ($\downarrow 0.4$) | $92.8 \pm 1.8$ ($\downarrow 0.8$) | $77.8 \pm 1.2$ ($\downarrow 1.0$) |
| | Quorus-Layerwise (5L) | $98.5 \pm 0.8$ | $97.5 \pm 0.4$ | $82.5 \pm 2.5$ ($\downarrow 1.2$) | $99.3 \pm 0.2$ | $92.4 \pm 2.6$ ($\downarrow 1.2$) | $78.8 \pm 1.1$ |
| | Quorus-Layerwise (8L) | $97.5 \pm 0.9$ ($\downarrow 1.0$) | $97.0 \pm 0.7$ ($\downarrow 0.5$) | $83.7 \pm 2.3$ | $98.8 \pm 0.9$ ($\downarrow 0.5$) | $93.6 \pm 0.5$ | $78.3 \pm 1.4$ ($\downarrow 0.5$) |
| 5 | Quorus-Layerwise (5L) | $98.3 \pm 0.9$ ($\downarrow 0.3$) | $97.3 \pm 0.8$ ($\downarrow 0.5$) | $81.7 \pm 4.4$ ($\downarrow 2.2$) | $98.8 \pm 0.3$ ($\downarrow 0.6$) | $92.9 \pm 1.8$ ($\downarrow 0.9$) | $77.7 \pm 1.0$ ($\downarrow 1.1$) |
| | Quorus-Layerwise (6L) | $98.6 \pm 0.8$ | $97.8 \pm 0.2$ | $83.1 \pm 2.4$ ($\downarrow 0.8$) | $99.4 \pm 0.3$ | $92.7 \pm 2.5$ ($\downarrow 1.1$) | $78.8 \pm 0.8$ |
| | Quorus-Layerwise (10L) | $97.5 \pm 0.8$ ($\downarrow 1.1$) | $97.0 \pm 0.6$ ($\downarrow 0.8$) | $83.9 \pm 2.2$ | $98.8 \pm 0.8$ ($\downarrow 0.6$) | $93.8 \pm 0.5$ | $78.1 \pm 1.5$ ($\downarrow 0.7$) |

for each layer) on IBM Kingston. We plot the accuracies of the classifiers of each layer, plotted in red dots, as well as the accuracies of the classifier ensemble up to that layer, plotted in yellow dots. We similarly plot the classifier accuracies from each layer in ideal simulation in dark blue dots, and the ensemble of the classifiers up to that layer in light blue.

One interesting observation is in the separation between the individual classifier and ensemble outputs in ideal and hardware evaluation. Notably, on IBM Kingston, although the testing accuracy is below 60% for individual classifiers for depth 3 and later, the ensemble of these classifiers generally increases, and maintains a nearly 20% separation at layer 5. This suggests that even though noise can corrupt individual classifier outputs, Quorus is robust to hardware errors from its in-built shot-efficient ensemble evaluation and is able to substantially mitigate these hardware errors.

# E  LLM USAGE

ChatGPT was used to help polish the writing of the work. All edits were subsequently verified by the authors. ChatGPT search was also used to find related work. All sources found were validated by the authors to be relevant.

# F  REPRODUCIBILITY STATEMENT

To ensure full reproducibility, we have released all code, datasets, and experiment configurations used in this work (also attached to this submission). The open-sourced repository includes detailed documentation, environment specifications, and ready-to-use scripts to replicate the results in the paper. By making these resources publicly available, we aim not only to guarantee transparency and independent verification of our results but also to accelerate future research by lowering barriers for benchmarking, extension, and adoption by the broader community.

**Repository Link:** `https://github.com/positivetechnologylab/Quorus`

# G  ABLATION: NON-IID DATASETS

We compare Quorus-Layerwise and Quorus-Funnel with Q-HeteroFL for Non-IID MNIST and Fashion-MNIST datas in Table 10. Similar to DepthFL (Kim et al., 2023), a *Dirichlet* distribution $\mathbf{p}_c \sim Dir_k(\beta = 0.5)$ was used to assign $p_{c,k}$ ratio of data samples of class $c$ to client $k$, with the constraint that each client has the same number of total training samples (128 samples). On average, the best performing Quorus variant has an 22.5% improvement in testing accuracy over Q-HeteroFL, with an 11.0% reduction in average standard deviation. In addition, Quorus-Layerwise and Quorus-Funnel have similar mean testing accuracies (often within 1%), highlighting their strength and interchangeability based on quantum hardware constraints. This table highlights not only the robustness of Quorus to non-IID data, but also its stability: for different data splits for Q-HeteroFL, the standard deviation is very high, often ranging from 10-20%. However, the standard

Table 10: Capacity-wise Comparison (V-Shape) — Q-HeteroFL vs. Quorus variants for Non-IID Data with $\Delta$ to the Best (**Bolded**). The means and standard deviations are calculated over five runs: Quorus-Layerwise and Quorus-Funnel outperform Q-HeteroFL with smaller standard deviations.

| Capacity | Technique | MNIST (Non-IID) | | | Fashion-MNIST (Non-IID) | | |
|---|---|---|---|---|---|---|---|
| | | 0/1 | 3/4 | 4/9 | Trouser/Boot | Bag/Sandal | Pullover/Coat |
| 2L | Q-HeteroFL | $89.1 \pm 8.8$ ($\downarrow$ 8.6) | $52.3 \pm 24.0$ ($\downarrow$ 42.4) | $60.0 \pm 4.4$ ($\downarrow$ 16.0) | $68.1 \pm 34.1$ ($\downarrow$ 30.2) | $70.2 \pm 14.9$ ($\downarrow$ 17.7) | $56.1 \pm 6.0$ ($\downarrow$ 15.5) |
| | Quorus-Layerwise | $\mathbf{97.7 \pm 0.8}$ | $94.6 \pm 3.3$ ($\downarrow$ 0.1) | $74.2 \pm 3.2$ ($\downarrow$ 1.8) | $\mathbf{98.3 \pm 1.0}$ | $87.9 \pm 3.1$ | $71.5 \pm 7.2$ ($\downarrow$ 0.1) |
| | Quorus-Funnel | $97.0 \pm 2.0$ ($\downarrow$ 0.7) | $\mathbf{94.7 \pm 1.5}$ | $\mathbf{76.0 \pm 4.1}$ | $98.0 \pm 1.2$ ($\downarrow$ 0.3) | $86.5 \pm 8.8$ ($\downarrow$ 1.4) | $\mathbf{71.6 \pm 4.8}$ |
| 3L | Q-HeteroFL | $77.0 \pm 11.0$ ($\downarrow$ 21.4) | $77.1 \pm 16.7$ ($\downarrow$ 19.1) | $57.3 \pm 4.6$ ($\downarrow$ 22.1) | $67.6 \pm 12.1$ ($\downarrow$ 31.5) | $62.1 \pm 12.3$ ($\downarrow$ 28.4) | $53.0 \pm 3.6$ ($\downarrow$ 23.1) |
| | Quorus-Layerwise | $\mathbf{98.4 \pm 0.5}$ | $95.1 \pm 3.5$ ($\downarrow$ 1.1) | $79.1 \pm 2.5$ ($\downarrow$ 0.3) | $\mathbf{99.1 \pm 0.4}$ | $90.5 \pm 2.6$ ($\downarrow$ 0.0) | $\mathbf{76.1 \pm 0.8}$ |
| | Quorus-Funnel | $97.8 \pm 1.5$ ($\downarrow$ 0.6) | $\mathbf{96.2 \pm 1.1}$ | $\mathbf{79.4 \pm 2.9}$ | $98.6 \pm 0.5$ ($\downarrow$ 0.5) | $\mathbf{90.5 \pm 1.9}$ | $75.0 \pm 3.2$ ($\downarrow$ 1.1) |
| 4L | Q-HeteroFL | $67.8 \pm 5.9$ ($\downarrow$ 30.6) | $73.9 \pm 17.9$ ($\downarrow$ 22.8) | $56.9 \pm 8.9$ ($\downarrow$ 25.3) | $74.7 \pm 19.7$ ($\downarrow$ 24.5) | $70.7 \pm 15.8$ ($\downarrow$ 20.8) | $53.6 \pm 4.2$ ($\downarrow$ 19.9) |
| | Quorus-Layerwise | $\mathbf{98.4 \pm 0.4}$ | $\mathbf{96.7 \pm 1.1}$ | $80.6 \pm 1.7$ ($\downarrow$ 1.6) | $\mathbf{99.2 \pm 0.4}$ | $\mathbf{91.5 \pm 3.0}$ | $\mathbf{73.5 \pm 4.2}$ |
| | Quorus-Funnel | $97.4 \pm 2.5$ ($\downarrow$ 1.0) | $96.5 \pm 0.9$ ($\downarrow$ 0.2) | $\mathbf{82.2 \pm 2.7}$ | $98.8 \pm 0.8$ ($\downarrow$ 0.4) | $90.0 \pm 3.4$ ($\downarrow$ 1.5) | $72.4 \pm 5.2$ ($\downarrow$ 1.1) |
| 5L | Q-HeteroFL | $80.6 \pm 12.6$ ($\downarrow$ 17.9) | $66.0 \pm 16.2$ ($\downarrow$ 31.2) | $57.7 \pm 5.8$ ($\downarrow$ 25.7) | $80.3 \pm 17.0$ ($\downarrow$ 18.9) | $72.8 \pm 18.0$ ($\downarrow$ 19.3) | $56.2 \pm 5.3$ ($\downarrow$ 17.6) |
| | Quorus-Layerwise | $\mathbf{98.5 \pm 0.4}$ | $\mathbf{97.2 \pm 0.6}$ | $81.6 \pm 1.4$ ($\downarrow$ 1.8) | $\mathbf{99.2 \pm 0.2}$ | $\mathbf{73.8 \pm 4.6}$ | $\mathbf{72.6 \pm 5.4}$ ($\downarrow$ 1.2) |
| | Quorus-Funnel | $97.6 \pm 2.1$ ($\downarrow$ 0.9) | $96.5 \pm 1.1$ ($\downarrow$ 0.7) | $\mathbf{83.4 \pm 2.5}$ | $98.7 \pm 1.0$ ($\downarrow$ 0.5) | $91.2 \pm 3.4$ ($\downarrow$ 0.9) | $72.6 \pm 5.4$ ($\downarrow$ 1.2) |
| 6L | Q-HeteroFL | $82.1 \pm 20.7$ ($\downarrow$ 16.6) | $74.4 \pm 12.7$ ($\downarrow$ 22.5) | $59.6 \pm 7.8$ ($\downarrow$ 22.7) | $79.1 \pm 17.8$ ($\downarrow$ 20.3) | $68.6 \pm 19.6$ ($\downarrow$ 23.6) | $57.0 \pm 7.9$ ($\downarrow$ 18.7) |
| | Quorus-Layerwise | $\mathbf{98.7 \pm 0.4}$ | $\mathbf{96.9 \pm 0.6}$ | $81.9 \pm 1.6$ ($\downarrow$ 0.4) | $\mathbf{99.4 \pm 0.1}$ | $\mathbf{92.2 \pm 1.9}$ | $\mathbf{75.7 \pm 2.7}$ |
| | Quorus-Funnel | $98.3 \pm 1.8$ ($\downarrow$ 0.4) | $96.1 \pm 1.5$ ($\downarrow$ 0.8) | $\mathbf{82.3 \pm 3.8}$ | $98.7 \pm 0.7$ ($\downarrow$ 0.7) | $89.8 \pm 8.3$ ($\downarrow$ 2.4) | $72.7 \pm 5.1$ ($\downarrow$ 3.0) |

deviation of Quorus-Layerwise and Quorus-Funnel never exceeds 10%, reflecting its stability across different non-IID datasets and its real-world utility.

# H  THEORETICAL ANALYSIS FOR ONE LAYER

Based on our results in Table 9, we provided empirical justification as to why we chose to use more than one layer of our V-shaped ansatz in the Quorus technique (because, with just one layer, Quorus performed significantly worse than when we used 2 layers). We provide a theoretical justification for why one layer is ineffective due to the measurement on the first qubit only depending on the $Z$ expectation on exactly half of the input qubits (and thus, only depending on half of the parameters in the first layer). Because of ineffective use of parameters, we suspect that the expressiveness of the circuit with only one layer is limited, motivating the need for additional layers for effective performance.

Thus, in this section, we provide a concrete theoretical analysis of the first layer behavior of our circuit. The decision function that we seek to learn from data depends on the parameters associated with the first qubit, which are reflected in the probability of obtaining the measurement outcomes $|0\rangle$ or $|1\rangle$. Ideally, we would like to express these measurement probabilities directly in terms of the amplitudes of the $N$-qubit product state produced after the initial single qubit rotations. This allows us to understand exactly how information from all input amplitudes flows into the measurement statistics of the first qubit under the given nearest neighbor CNOT architecture.

**Lemma 1** (First-qubit measurement after two CNOT sweeps). *Let $N \geq 2$ qubits be arranged linearly and initialized in a product state*

$$\rho_{\text{in}} = \bigotimes_{j=1}^{N} \rho_j.$$

*Let $U$ be the unitary corresponding to two layers of nearest-neighbor CNOT gates:*

- *a rightward sweep: CNOT$(1{\rightarrow}2)$, CNOT$(2{\rightarrow}3)$, ..., CNOT$(N{-}1{\rightarrow}N)$;*

- *followed by a leftward sweep: CNOT$(N \rightarrow N{-}1)$, CNOT$(N{-}1 \rightarrow N{-}2)$, ..., CNOT$(2 \rightarrow 1)$.*

*Then the Heisenberg-evolved observable of the first qubit satisfies*

$$U^\dagger Z_1 U = \prod_{\substack{1 \leq j \leq N \\ j \equiv N \,(\mathrm{mod}\, 2)}} Z_j.$$

*In particular, if*

$$z_j := \mathrm{Tr}(Z \, \rho_j) \in [-1, 1]$$

*denotes the Bloch $Z$-component of the $j$th input qubit, then the measurement probabilities of the first qubit in the $Z$ basis are*

$$\Pr(\textit{first qubit} = 0) = \frac{1}{2}\left(1 + \prod_{\substack{1 \le j \le N \\ j \equiv N \,(\mathrm{mod}\,2)}} z_j\right), \qquad \Pr(\textit{first qubit} = 1) = \frac{1}{2}\left(1 - \prod_{\substack{1 \le j \le N \\ j \equiv N \,(\mathrm{mod}\,2)}} z_j\right).$$

*In other words, the first-qubit measurement statistics depend only on the $Z$-components of those input qubits whose indices have the same parity as $N$ (odd indices if $N$ is odd, even indices if $N$ is even).*

*Proof.* First consider a computational-basis input $|x_1 x_2 \ldots x_N\rangle$ with $x_j \in \{0, 1\}$. A CNOT with control bit $a$ and target bit $b$ acts as $(a, b) \mapsto (a, a \oplus b)$, where $\oplus$ denotes XOR. In the rightward sweep $1 \to 2, 2 \to 3, \ldots, (N-1) \to N$, a straightforward induction shows that the output bits $y_1, \ldots, y_N$ satisfy

$$y_k = x_1 \oplus x_2 \oplus \cdots \oplus x_k, \qquad k = 1, \ldots, N.$$

In the subsequent leftward sweep $N \to N-1, \ldots, 2 \to 1$, each $\mathrm{CNOT}(i+1 \to i)$ updates $(y_i, y_{i+1})$ as $(y_i, y_{i+1}) \mapsto (y_i \oplus y_{i+1}, y_{i+1})$. The final first bit $x_1'$ can therefore be written as

$$x_1' = y_1 \oplus y_2 \oplus \cdots \oplus y_N.$$

Substituting $y_k = x_1 \oplus \cdots \oplus x_k$ and counting how many times each $x_j$ appears, we find that $x_j$ is included exactly $N - j + 1$ times, so only those $x_j$ with $(N - j + 1) \bmod 2 = 1$ contribute. Equivalently, $j \equiv N \pmod 2$, and hence

$$x_1' = \bigoplus_{\substack{1 \le j \le N \\ j \equiv N \,(\mathrm{mod}\,2)}} x_j.$$

For Clifford circuits, this classical XOR mapping of computational-basis labels coincides with the Heisenberg evolution of Pauli $Z$ operators Gottesman (1998). Equivalently, one may propagate $Z_1$ backwards through the circuit using the CNOT conjugation rules

$$\mathrm{CNOT}(c \to t)^\dagger \, Z_c \, \mathrm{CNOT}(c \to t) = Z_c, \qquad \mathrm{CNOT}(c \to t)^\dagger \, Z_t \, \mathrm{CNOT}(c \to t) = Z_c Z_t,$$

which yields the same expression

$$U^\dagger Z_1 U = \prod_{\substack{1 \le j \le N \\ j \equiv N \,(\mathrm{mod}\,2)}} Z_j.$$

For a product input state $\rho_{\mathrm{in}} = \bigotimes_{j=1}^N \rho_j$, the expectation of this operator factorizes:

$$\langle U^\dagger Z_1 U \rangle = \prod_{\substack{1 \le j \le N \\ j \equiv N \,(\mathrm{mod}\,2)}} \mathrm{Tr}(Z_j \, \rho_j) = \prod_{\substack{1 \le j \le N \\ j \equiv N \,(\mathrm{mod}\,2)}} z_j.$$

Finally, measuring the first qubit in the $Z$ basis uses the projectors $\Pi_0 = (I + Z_1)/2$ and $\Pi_1 = (I - Z_1)/2$, giving

$$\Pr(\text{first qubit} = 0) = \frac{1 + \langle U^\dagger Z_1 U \rangle}{2}, \qquad \Pr(\text{first qubit} = 1) = \frac{1 - \langle U^\dagger Z_1 U \rangle}{2},$$

which yields the claimed formulas upon substitution. $\qquad\square$

The implications of this analysis are that, because the probability of measuring the first qubit to be 0 or 1 only depends on half of the qubits' Z-expectation, half of the parameter information is not being used if we measure immediately after the first layer, which may be a reason why Quorus performs badly with just one layer – not enough parameters contribute to the loss function. This motivates the use of additional layers in Quorus so that our loss function is dependent on more parameters.

# I    COSTS OF USING QUORUS

We would like to provide a more detailed discussion of the costs of using Quorus. We will break down our discussion into two sections: classical resource cost, and quantum resource cost. Within the quantum resource cost, we will discuss the cost of using each of our design more explicitly. To our knowledge, a detailed discussion of how to compute layerwise losses in QML has not been proposed, which is what we provide below.

## I.1    CLASSICAL RESOURCE COST

The classical cost of using Quorus can be characterized as follows:

1. Client storage of parameters, model output, and gradient information
2. Server parameter aggregation process
3. Parameters sent over classical channel (network)

**One important observation is that model inference and gradient computation has minimal classical overhead**. The reason for this is that weight matrix multiplication and gradients can be computed natively on quantum hardware  (Wierichs et al., 2022), meaning that clients do not need to store large weight matrices nor the intermediate activations from each layer. Typically, storing these weight matrices as well as intermediate activations for gradient computation is a source of significant classical memory overhead, serving as the bottleneck for why clients with little memory cannot run large models. **As this overhead does not exist in quantum models due to quantum-native calculations, the bottleneck for clients is not limited classical memory, but rather limited depth quantum computation**.

The overhead that Quorus incurs (listed above) is small (on the order of the parameter count for each client), which is not expected to be a classical memory overhead for our quantum clients.

## I.2    QUANTUM RESOURCE COST

In contrast to the relatively small classical resource cost that Quorus incurs, the quantum resource cost that Quorus incurs needs to be carefully analyzed. As discussed in the main text, quantum computers are very expensive to run today, so minimizing the number of shots on the quantum computer is essential for the practicality of Quorus. **We will argue that, because Quorus-Layerwise incurs a shot overhead, we propose Quorus-Ancilla, Quorus-Blocking, and Quorus-Funnel, tailored for clients' hardware constraints to enable practical QFL with Quorus.**

### I.2.1    QUORUS-LAYERWISE OVERHEAD

**Quorus-Layerwise incurs a factor of $L$ overhead in terms of shot-count because the circuit must be re-run for each layer in order to compute the loss function**, where $L$ is the number of layers in the circuit. To understand this, we will walk through the steps a client must take to evaluate its loss function (which we defined in Equation 1). First of all, note that $L_{ce}^i(p_i, y)$ and $D_{KL}(p_j||p_i)$ both require the term $p_i$, which are the logits output from the i-th layer. In our layerwise loss function, we see the expressions $\sum_{i=1}^{d_k} L_{ce}(p_i)$ and $\sum_{i=1}^{d_k} \sum_{j=1, j \neq i}^{d_k} D_{KL}(p_j||p_i)$. Note that the outer sum of these expressions is both $\sum_{i=1}^{d_k}$. This means that the client must be able to obtain $p_i$ for each $1 \leq i \leq d_k$ to compute its loss $L_k$. Obtaining those $p_i$'s is precisely what we focus on.

In this Quorus-Layerwise approach, we propose simply rerunning the circuit for each depth (as described in Subsec. 4.3). To be more precise, for each depth $i$ from $1, 2, ..., d_k$, we have to rerun the circuit. The reason we do this is to preserve the exact state fed into the next layer of the quantum circuit, which is not possible with a midcircuit measurement (for detailed discussion of why this is the case, refer to Subsec. 4.3).

Then, if we need $S$ shots to obtain the probability distribution $p_i$ for our quantum model, then we will need $SL$ shots to obtain all $p_i, 1 \leq i \leq d_k$. This is precisely the linear factor in shot-overhead.

And the fact that this is a multiplicative factor is meaningful, once we consider how the total shot-count scales when we do gradient computation according to the parameter-shift rule (Wierichs et al., 2022). To analyze this, we will employ a counting argument.

**Proposition 2** (Quadratic shot count with Quorus–Layerwise). *Let $n$ be the number of qubits and $L$ the number of layers in the ansatz of Fig. 2. Assume that, for each layer index $i \in \{1, \ldots, L\}$, the scalar quantity $p_i$ can be estimated with $S$ shots, independently of $i$ and of the parameters.*

*Then, using Quorus–Layerwise and the parameter-shift rule, the total number of shots needed to compute the gradient of $L_k$ with respect to* all *parameters of the ansatz is*

$$T_{\text{layerwise}} = 6Sn \sum_{\ell=1}^{L} (L - \ell + 1) = 3SnL(L+1),$$

*which scales quadratically in $L$.*

*Moreover, if the loss $L_k$ for client $k$ can be evaluated with $S$ shots (i.e., independently of the layer index $\ell$), then the total number of shots needed to compute the gradients with respect to all parameters is*

$$T_{\text{const}} = 6SnL,$$

*which scales linearly in $L$.*

*Proof.* For the ansatz in Fig. 2, we index the parameters as

$$\theta = (\theta_{q,\ell,r})_{q,\ell,r} \in \mathbb{R}^{n \times L \times 3},$$

where

- $q \in \{1, \ldots, n\}$ denotes the qubit,

- $\ell \in \{1, \ldots, L\}$ denotes the layer,

- $r \in \{1, 2, 3\}$ denotes the position of the parameter in the single-qubit rotation gate.

On each qubit $q$ in each layer $\ell$ we apply a three-parameter gate

$$Rot(\theta_{q,\ell,1}, \theta_{q,\ell,2}, \theta_{q,\ell,3}) := R_z(\theta_{q,\ell,3}) \, R_y(\theta_{q,\ell,2}) \, R_z(\theta_{q,\ell,1}),$$

as defined in Appendix A. Thus, each layer contains $n$ such gates and therefore $3n$ parameters.

We analyze the cost of computing the gradient with respect to a fixed parameter $\theta_{q,\ell,r} \in \mathbb{R}$, i.e., $\frac{\partial L_k}{\partial \theta_{q,\ell,r}}$. By the parameter-shift rule Wierichs et al. (2022), this gradient can be written as

$$\frac{\partial L_k}{\partial \theta_{q,\ell,r}} \propto L_k(\theta^{+(q,\ell,r)}) - L_k(\theta^{-(q,\ell,r)}),$$

where, for some fixed shift $\varepsilon \in \mathbb{R}$,

$$\theta^{\pm(q,\ell,r)} := \theta \pm \varepsilon \, e^{(q,\ell,r)},$$

and $e^{(q,\ell,r)} \in \mathbb{R}^{n \times L \times 3}$ is the basis tensor with entries

$$e_{a,b,c}^{(q,\ell,r)} = \begin{cases} 1, & \text{if } (a,b,c) = (q,\ell,r), \\ 0, & \text{otherwise.} \end{cases}$$

Thus, it suffices to count the number of shots required to evaluate $L_k(\theta^{\pm(q,\ell,r)})$ for this parameter. Recall that $L_k$ (Eq. 1) is defined in terms of the layerwise quantities $p_i$. For a parameter in layer $\ell$, the corresponding circuit only affects the outputs $p_i$ with $i \geq \ell$. Therefore:

- for $j < \ell$, the parameter $\theta_{q,\ell,r}$ does not appear in the circuit used to compute $p_j$, so $p_j$ is independent of $\theta_{q,\ell,r}$ and $\frac{\partial L_{ce}^j}{\partial \theta_{q,\ell,r}} = 0$;

- for all $i$ with $\ell \leq i \leq d_k$ (or up to $L$ in the worst case), the parameter $\theta_{q,\ell,r}$ may influence $p_i$, so we must run the circuit to estimate $p_i$.

Under the assumption of the proposition, evaluating $p_i$ for a given $i$ costs $S$ shots. Therefore, to compute $L_k$ for a parameter in layer $\ell$, we need to evaluate $p_i$ for $i = \ell, \ldots, L$, which requires

$$S(L - \ell + 1)$$

shots for a single evaluation of $L_k$. Because the parameter-shift rule requires two such evaluations, $L_k(\theta^{+(q,\ell,r)})$ and $L_k(\theta^{-(q,\ell,r)})$, the cost for the single parameter $\theta_{q,\ell,r}$ is

$$T_{q,\ell,r} = 2S(L - \ell + 1).$$

Each layer $\ell$ has $3n$ parameters (three per gate, one gate per qubit), and $T_{q,\ell,r}$ depends only on $\ell$. Hence, the total number of shots for all parameters in layer $\ell$ is

$$T_\ell = 3n \cdot T_{q,\ell,r} = 6Sn(L - \ell + 1).$$

Summing over all layers yields the total shot count:

$$T_{\text{layerwise}} = \sum_{\ell=1}^{L} T_\ell = \sum_{\ell=1}^{L} 6Sn(L - \ell + 1)$$

$$= 6Sn \sum_{\ell=1}^{L} (L - \ell + 1) = 6Sn \sum_{t=1}^{L} t$$

$$= 6Sn \frac{L(L+1)}{2} = 3SnL(L+1),$$

which proves the first claim.

For the second claim, assume instead that the full loss $L_k$ (for any parameter setting $\theta$ and any layer index) can be evaluated with $S$ shots. Then, for each parameter $\theta_{q,\ell,r}$, the parameter-shift rule requires two evaluations of $L_k$, costing $2S$ shots. Since there are $3nL$ parameters in total, the shot count is

$$T_{\text{const}} = 3nL \cdot 2S = 6SnL,$$

as claimed. $\qquad\square$

From Proposition 2, we see that Quorus-Layerwise has a quadratic shot overhead compared to a loss function that can be evaluated with just $S$ shots (which is standard in variational quantum algorithms). To reduce this shot overhead, we aim to evaluate our loss function $L_k$ in just $S$ shots, independent of the layer count $L$. This will be detailed next.

### I.2.2 QUORUS-ANCILLA OVERHEAD

To that end, we propose a technique, Quorus-Ancilla, that uses ancilla(e) qubit(s). As depicted in Fig. 5, we entangle the first qubit with an ancilla qubit after each layer. **Thus, if the client has $L$ layers for $C$ class classification, then we require the client to have $L \times \lceil log_2(C) \rceil$ ancilla qubits, with the ability to do a two-qubit gate from the first $\lceil log_2(C) \rceil$ qubits to each ancilla**. Note that, in doing so, the loss function $L_k$ for client $k$ can now be computed in just $S$ shots. To do so, the client simply needs to obtain the marginal probability distribution over $\lceil log_2(C) \rceil$ ancillae of each of the $L$ groups of $\lceil log_2(C) \rceil$ ancillae; for group of $\lceil log_2(C) \rceil$ ancillae qubits $a_i$, this will yield the marginal distribution $p_i$. With just $S$ shots of the circuit, each group of ancillae $a_i$ will be measured $S$ times, which gives us $S$ values for each group of ancillae to form the histogram that we use as our estimate of $p_i$. With $p_i$, we can compute our loss function $L_k$ with classical postprocessing.

We would like to highlight how the hardware constraint of additional ancillae and higher connectivity is amenable for some quantum hardware architectures but not others. For example, superconducting machines have limited connectivity and cannot natively perform two-qubit gates between one qubit and an arbitrary number of other qubits without additional overhead (Lange et al., 2025).

However, trapped-ion machines exhibit all-to-all connectivity, and neutral atom machines are reconfigurable and can, in theory, support two-qubit gates between any qubits (Chen et al., 2024; Evered et al., 2023). Thus, for trapped-ion and neutral atom clients that are resource-constrained (cannot run many shots), Quorus-Ancilla is amenable to their hardware capabilities.

Finally, we would like note that the user actually does not need to have $L$ ancillae, provided they have some way to reset the qubit mid-computation. Using just $\lceil log_2(C) \rceil$ ancilla qubits and resetting their state to 0 after each layer would also be sufficient to implement the Quorus-Ancilla approach. However, if the measurement and reset operation takes a long time, then that could block the operations in the circuit's next layer and induce latency in the circuit. Thus, using $\lceil log_2(C) \rceil$ ancilla qubits is particularly amenable to clients who have the ability to reset a qubit in parallel with other circuit operations, with real-time cost that is similar to that of the circuit operations. This hardware requirement is also used in quantum error correction (DeCross et al., 2022). Thus, as quantum hardware tailored for quantum error correction improves, the same functionality can be used to support this technique.

### I.2.3    QUORUS-BLOCKING OVERHEAD

For a client that does not have additional ancilla qubits, we propose an alternate version of Quorus, called Quorus-Blocking. Depicted in Fig. 6 (a), **we require the client to have the ability to do a mid-circuit measurement on the first** $\lceil log_2(C) \rceil$ **qubits after each layer**. We store the measurement outcome on the $\lceil log_2(C) \rceil$ ancillae at each layer for each shot; thus, with $S$ shots, we will have $S$ values for each layer to form the histogram which we use as an estimator for $p_i$.

We would like to highlight that for clients with the ability to implement mid-circuit measurements quickly, Quorus-Blocking is particularly amenable. And, as discussed above, with quantum hardware built to support quantum error correction, the same capability of doing fast mid-circuit measurements will enable the practicality of Quorus-Blocking.

### I.2.4    QUORUS-FUNNEL OVERHEAD

Despite the real-world relevance of the hardware requirements discussed in the above sections, there is quantum hardware today where none of the above constraints are satisfied, given the relative nascency of quantum hardware. To support clients with the least amount of hardware constraints, we trade-off quantum hardware constraints with quantum model size. **For the Quorus-Funnel approach, we require the client to use a model that acts on** $\lceil log_2(C) \rceil$ **fewer qubits after each layer.** The reason for this is, if $\lceil log_2(C) \rceil$ qubits are not used for computation in any further layer, then they can be measured without blocking other computation. Thus, by obtaining the marginal distribution over the first $\lceil log_2(C) \rceil$ remaining qubits after each layer, with just $S$ shots, we can obtain an estimate for $p_i$ for each layer.

### I.3    SUMMARY OF QUORUS RESOURCE COSTS

In conclusion, we have discussed why Quorus does not incur a high classical resource cost over other QFL methods, and discussed the unique and practically relevant hardware requirements that each version of Quorus has. In doing so, we highlight the relevance of Quorus to quantum platforms in our current-day and ones we expect in the near future.

