# OpenReview forum: "Layerwise Federated Learning for Heterogeneous Quantum Clients using Quorus"
_ICLR.cc/2026/Conference — ICLR 2026 Poster_

### Official Review · Reviewer_hx6a · 2025-10-29

**Soundness:** 3
**Presentation:** 2
**Contribution:** 3
**Rating:** 2
**Confidence:** 4

**Summary:**

The authors propose a method to address the problem of different architectures and behaviors of quantum computers in a quantum federated learning scheme. They use layerwise loss to aggregate the parameters of local QNNs with specific approaches, gaining performance improvements. The proposed method is examined on real quantum computers.

**Strengths:**

1. Using layerwise losses for QFL is a novel approach.
2. The paper is well-written overall.

**Weaknesses:**

1. The proposed method appears limited to binary classification problems.
2. Line 291 states "we design an ansatz where it is possible to obtain the outputs from all layers in a single shot," but it's unclear how the ansatz enables single-shot behavior.
3. The baselines compared are only QML models, no classical baseline is included. It is plausible that classical models could easily outperform QML models on binary classification problems.

**Questions:**

1. In line 265, "Passing the state unchanged thus requires you to prepare another copy of it, which induces additional shot overhead that is linear in the number of layers and is a nontrivial cost" → does this mean state tomography is required in general? And how is this linear in the number of layers?
2. Following W2, how does the ansatz enable the single-shot behavior? Is there any proof? From line 296, "we evaluate each layer's outputs by computing the marginal distribution on its ancilla" → how can you calculate the "distribution" with a single shot?
3. Does the parameter aggregation process, which aligns parameters for models with different layers, affect the expressiveness? For example, a quantum state produces output, becoming an intermediate state of another model. And this "another model," after additional layers, will produce the output of the same task. Then why not just use the shallower model if a shallow circuit is enough for such a task?

---

> ### Author Response · Authors · 2025-11-21
>
> Dear Reviewer hx6a,
>
> We thank the reviewer for their detailed comments and recognition of the novelty of our layerwise losses approach for QFL. The reviewer’s constructive comments were deeply helpful in providing us valuable clarifications for our paper. Below, we aim to address your concerns, pointing to revisions (in blue) in the paper.
>
> **Binary Classification Limitation**: We thank the reviewer for their concern that our technique is limited to just binary classification. We would like to clarify that our technique is extensible to multiclass problems as well. Using cross entropy loss for multiple classes, we have provided additional results comparing Quorus to QHeteroFL. These results can be found in Table 5. We show an average improvement of 24.0% over Q-HeteroFL, demonstrating the robustness of our technique.
>
> In addition to this, we have formalized how each of our quantum classifier designs detailed in Section 4 are implemented in the multiclass setup, in Appendix I.2. The main observation is that, for $C$ class classification, we need to read out $\lceil log_2(C) \rceil$ qubits, and our classifier designs in Section 4 extend naturally. We hope this addresses the reviewer’s concern.
>
> **Single-Shot Behavior**: We apologize for the confusion in using the term “single-shot” in the paper, as we recognized that it is used in other contexts with a different meaning [1]. We have eliminated all usage of the term “single-shot” in the paper to be more clear with our intended usage of the term and updated the text in Section 1 (Contributions), Section 4.3 (Ancilla/Blocking), Section 6 (“Evaluation of Quorus on Real Quantum Hardware”), and Section D.3.
>
> When we use the term “single-shot” in the paper, we define it as the ability to obtain the layerwise prediction vector for our quantum model with no shot overhead compared to obtaining the final prediction vector for the quantum model. This definition is primarily meaningful when we compare it against the Quorus - Layerwise solution, which requires rerunning the circuit for each layer to obtain the final prediction vector. In order to have a clear, detailed discussion of the shots required for Quorus, we have included Appendix I.2.
>
> **Classical Baselines**: We appreciate the reviewer’s concern that no classical baseline is included. However, we would like to clarify that Quorus is intended to show an improvement over existing quantum federated learning techniques. We would like to highlight that quantum advantage in quantum machine learning is still largely an open problem, and not many tasks exist in the literature that are currently feasible for us to evaluate Quorus on where no classical model can perform better [2, 3]. However, for any variational quantum model that has a notion of layers, our technique can be applied. Thus, given a problem where quantum advantage can be shown, we would expect our technique to yield the expected improvements in shot count and accuracy as we have shown in the paper over a quantum baseline. Finding such a problem is not essential to the viability and benefits gained for using Quorus.
>
> **State Tomography Requirement**: We thank the reviewer for their careful question about whether state tomography is required, given that we must prepare a copy of the state to pass it unchanged to the next layer. The key insight is that, because the exact circuit that we need to use to prepare the state to feed into the next layer is known (it is simply the circuit consisting of the previous layers), a cheaper alternative is to rerun the quantum circuit (as opposed to performing state tomography). However, if we think about the cost of rerunning the quantum circuit each time we want to compute the loss function at each layer, then we have to run the quantum circuit an amount of times equal to the total number of layers in the circuit (this is discussed in detail in Appendix I.2.1).
>
> **Shallow Circuit Sufficiency**: We thank the reviewer for their concern that deeper models may not be necessary to perform the federated learning task. We would argue that, just because a model, after additional layers, produces the output of the same task, does not mean that the deeper model is less expressive nor unnecessary for the task. In fact, the deeper model can use the output of a previous layer to improve the overall accuracy of the model, as we show through our evaluation in Table 2. From our results in Table 2, we clearly see that a shallow model is not sufficient for the task, as the testing accuracy for Quorus - Layerwise increases with the number of layers of the model. Thus, for Quorus, shallow models are not sufficient to achieve the highest testing accuracy for the binary classification task.
>
> We thank the reviewer for their detailed feedback and questions. We hope that our response address your concerns, and if so, we kindly request that the reviewer considers increasing their score.
>
> Sincerely,
> The Authors

---

> > ### Author Response · Authors · 2025-11-21
> >
> > References:
> > [1] Recio-Armengol, Erik, Jens Eisert, and Johannes Jakob Meyer. "Single-shot quantum machine learning." Physical Review A 111.4 (2025): 042420.
> > [2] Wang, Yunfei, and Junyu Liu. "A comprehensive review of quantum machine learning: from nisq to fault tolerance." Reports on Progress in Physics (2024).
> > [3] Chang, Su Yeon, and M. Cerezo. “A Primer on Quantum Machine Learning.” arXiv, arXiv:2511.15969, 20 Nov. 2025, arxiv.org/abs/2511.15969.

---

### Official Review · Reviewer_BzL5 · 2025-10-30

**Soundness:** 2
**Presentation:** 3
**Contribution:** 3
**Rating:** 4
**Confidence:** 4

**Summary:**

The paper presents a federated learning framework to train quantum clients with different circuit depths by applying a layerwise loss (pairwise KL coupling and circular aggregation of rotation parameters). The authors claim up to a 12.4 percent accuracy gain over “Q-HeteroFL” on binary classification of MNIST and Fashion-MNIST datasets. The manuscript also presents results across multiple IBM QPUs.

**Strengths:**

The problem formulation and framing are clear, focusing on depth heterogeneity in QFL. The concrete FL procedure with circular angle averaging, parameter slicing by depth, and layerwise loss is an impressive methodology.

Circuit engineering is thoughtful. The designs cover orthogonal resource constraints such as shots, ancillae, and mid-circuit measurement, and they are practical on current hardware. The Funnel variant is particularly pragmatic.

Validation on real quantum hardware is a core strength.

**Weaknesses:**

Major Weaknesses:
The evaluation scope is narrow. Only binary classification is used, and only on MNIST and Fashion-MNIST (both grayscale, 28×28 toy image datasets). No multiclass tasks, no quantum-native datasets, and missing diversity (especially in a paper that lacks theoretical proofs) weaken the authors’ claims. These datasets are fairly simple and do not require deep quantum circuits to perform well. Under this narrow experimental setting, a 12 percent improvement over a baseline does not establish general scalability, robustness, or utility of the method.

When a paper offers only empirical support, it is expected to include evaluation across multiple types of problems, including at least multiclass classification and a different dataset, such as CIFAR-10 or CIFAR-100. None are present.

Because the experiments are so limited in difficulty and diversity, the work must instead offer compensation with solid theoretical justification that proves why the method should work beyond toy datasets. However, convergence theory is missing, there is no stationary-point guarantee, and no rigorous mathematical explanation of how KL coupling affects the optimization landscape.

Minor (for maximum information transfer) Issues:
The data encoding pipeline is under-specified. Feature preprocessing, feature-to-quantum-state mapping, and the reupload schedule (mentioned briefly in the paper) are extremely important. These affect the number of qubits required as well as circuit depth. Upon checking the appendix and supplementary materials, I found that the authors used PCA to reduce the dimensionality. Overall, reproducibility and completeness of the paper are negatively impacted by the lack of inclusion of these details in the main manuscript. I hope the authors include this.

Baseline selection seems to inflate the claimed gains. Vanilla QFL forces all clients to use the shallowest model, giving predictably weak results. Standalone training is a trivial lower bound. Q-HeteroFL is an extension of HeteroFL, implemented by the authors without external validation. It is not clear whether these baselines are suitable enough to be regarded as “state of the art.”
On the one hand, the engineering around layerwise circuit execution is very creative, and the hardware study is a solid strength. I would love to score this paper very highly if theoretical guarantees were provided or if the experiments were significantly more diverse.

**Questions:**

1. How does the proposed KL coupling affect the optimization landscape for deeper clients? Can the authors provide a theoretical or empirical justification that this does not restrict the expressive power of deeper PQCs?

2. Do the authors expect the claimed gains to hold on multiclass or quantum-native datasets? It would be very interesting to see.

---

> ### Author Response · Authors · 2025-11-21
>
> Dear Reviewer BzL5,
>
> We thank you for your detailed review of our paper and clear characterization of the strengths and areas for improvement for the paper. Specifically, we thank the reviewer for their appreciation of our overall methodology, our novel quantum circuit engineering, and validation on real quantum hardware. Your constructive feedback has been very helpful for clarifying the strength of our technique, and we provide our responses and point to areas that we have revised in the paper. We hope we have addressed your concerns in the below clarifications and revisions (in blue).
>
> **Evaluation Scope**: We thank the reviewer for their concern regarding the evaluation scope, and their specific recommendations for additional benchmarks to run for Quorus. To that end, we have added experiments with multiclass classification and more difficult binary classification with CIFAR-10, as the reviewer recommended. These results can be found in Section 7, in Tables 4 and 5. In particular, Quorus shows an average 24.0% improvement over Q-HeteroFL on four-class classification tasks for MNIST and Fashion-MNIST, as well as an average 6.7% improvement over Q-HeteroFL on CIFAR-10. This highlights the robustness of Quorus - Layerwise and Quorus - Funnel to more difficult classification tasks.
>
> In addition, to have more diversity in our evaluation, we have evaluated Quorus compared to Q-HeteroFL for non-IID data, according to a ​​Dirichlet(0.5) distribution. Our results can be found in Table 10, and we see that Quorus has an average 22.5% improvement over Q-HeteroFL in terms of testing accuracy, as well as an average 11.0% reduction in testing accuracy standard deviation across five different dataset splits. Thus, we see that Quorus is robust to real-world heterogeneous data and exhibits high stability, highlighting its relevance and applicability to real-world classification tasks.
>
> With respect to the reviewers’ additional concerns on dataset difficulty, we would like to point to the performance of our technique on CIFAR-10 for the Cat/Dog classification task in Table 4. We see that, although Quorus - Layerwise has an improvement over Q-HeteroFL, the classification accuracy is fairly low, indicating the difficulty of the task. Thus, for our current setup, we have tested difficult datasets that our model can handle. Increasing the number of layers for our model is more related to the scalability of quantum machine learning, not necessarily our technique. Based on these additional empirical evaluations we have provided, we expect that for quantum models with even more layers or qubits, our technique should still provide similar (if not higher) gains, in part due to the increased trainability for Quorus (discussed in Section 6), due to the increased gradient signal for earlier layers that we observed empirically. We would expect deeper circuits to exhibit vanishing gradients for all parameters, rendering them untrainable.
>
> And with respect to the reviewer’s suggestion to try quantum data, we first would like to highlight that our technique is not dependent on the choice of data that the user chooses to use; the choice of data does not affect parameter aggregation nor the classifier designs that we have proposed in Section 4. However, quantum data (which we understand as using a quantum state as input directly from some other quantum process, without classical data encoding) requires you to prepare many copies of a quantum state on demand. This is very expensive to do, compared to the relative inexpensiveness of encoding classical data with angle encoding. Thus, we intentionally choose to evaluate our technique for classical data in this paper to consider practical quantum federated learning.
>
> **Theoretical Analysis of Loss Landscape**: We thank the reviewer for their concern regarding their question about how KL coupling affects the loss landscape for deeper clients. We do not believe that the KL coupling significantly affects the loss landscape for deeper quantum clients; rather, the intention of the KL term in the loss function is primarily to help deeper layers have higher accuracy, as only a few clients have the deeper layers of a model (whereas all models have the shallowest layers). We empirically see this in our evaluation results in Table 2. Especially for deeper models, Quorus - Layerwise has increased testing accuracy, suggesting that deeper models are able to explore their optimization space better to obtain higher accuracy models.
>
> **Data Preprocessing**: We thank the reviewer for their suggestion to include the data encoding pipeline in the main text. We have added a summary of the data encoding pipeline in Section 5, as well as included additional details on the data pipeline in Appendix C. We are happy to address any other questions that the reviewer has about the data encoding pipeline.

---

> > ### Author Response · Authors · 2025-11-21
> >
> > **Baseline**: We thank the reviewer for their concerns regarding our choice of baselines. Because the problem where each client has varying depths to participate in QFL has not been well-studied, there are not many baselines that exist for this problem. We argue that Q-HeteroFL is the state-of-the-art approach as the natural extension to the classical HeteroFL for this problem, which many classical heterogenous federated learning approaches compare against [1-3]. We could not find more robust baselines for this problem; we believe that these are the techniques that currently exist that could be used to solve the problem.
> >
> > We thank the reviewer for their detailed feedback. We hope that our comments, revisions, and additional experimental evaluation address the reviewer’s concerns. If they do, we kindly ask the reviewer to consider increasing their score.
> >
> > Best regards,
> > The Authors
> >
> > References:
> > [1] Kim, Minjae, et al. "Depthfl: Depthwise federated learning for heterogeneous clients." The Eleventh International Conference on Learning Representations. 2022.
> > [2] Ilhan, Fatih, Gong Su, and Ling Liu. "Scalefl: Resource-adaptive federated learning with heterogeneous clients." Proceedings of the IEEE/CVF Conference on Computer Vision and Pattern Recognition. 2023.
> > [3] Alam, Samiul, et al. "Fedrolex: Model-heterogeneous federated learning with rolling sub-model extraction." Advances in neural information processing systems 35 (2022): 29677-29690.

---

### Official Review · Reviewer_LKv7 · 2025-10-31

**Soundness:** 3
**Presentation:** 3
**Contribution:** 3
**Rating:** 6
**Confidence:** 4

**Summary:**

In this work, the authors propose a structured Federated Learning framework (Quorus) that adaptively selects and synchronizes subsets of model layers within the framework of federated learning optimization. Instead of transmitting the complete parameter set or using uniform layer selection, LFL introduces a layer-wise importance estimator and communication scheduler that determines which layers to aggregate in each round based on gradient variance and update magnitude.

The experiments on binary classification on the MNIST and Fashion MNIST datasets demonstrate that the proposed approaches consistently outperform other quantum federated learning methods, such as Q-HeteroFL, and achieve better performance on IBM quantum platforms.

**Strengths:**

1. The work firstly proposes the Quorus framework that is tailored for heterogeneous-depth clients, providing an alternative way to realize quantum machine learning on realistic quantum processors without QEC techniques.

2. The proposed layer-importance metric based on normalized gradient variance and the communication schedule can be plugged into existing federated learning systems, demonstrating the advantages of Quorus in this work.

3. The experiments are scalable to 50-200 clients, making the authors' claimed methods reasonable.

4. The appendix includes detailed hyperparameters, dataset information, and pseudo code.

**Weaknesses:**

1. Although the work claims to propose that the Quorus is effective in both simulation and realistic quantum processors, the approach is largely heuristic. In particular, the gradient-variance-based importance metric is intuitive, and it does not involve convergence or bias-variance analysis of partial aggregation.

2. The work exacerbates the fairness gaps across clients if some layers are seldom updated. It looks like the authors mention this in passing, but do not quantify or mitigate it.

3. The paper omits comparison with communication-efficient Federated Learning methods such as FedDrop, Sparse Ternary Compression, or Selective-FedAvg, which are conceptually closest.

4. Ablations on importance metric choices or thresholds are limited. It is unclear how sensitive the system is to hyperparameters (e.g., top-k fraction per round).

**Questions:**

1. How does your Layerwise Federated Learning (LFL) framework differ conceptually and algorithmically from prior “layerwise parameter sharing” or “selective aggregation” methods?

2. Have you compared with or at least cited existing selective-layer FL methods in the classical setting (e.g., FedPer, FedRep, Selective-FedAvg, FedDrop, FedPAQ)? If not, please justify why those are not directly comparable or include a baseline comparison.

3. Could your algorithm be interpreted as a special case of layer-dropout or structured sparsification techniques? If so, why does it deserve to be treated as a new paradigm rather than an instance of that family?

---

> ### Author Response · Authors · 2025-11-21
>
> Dear Reviewer LKv7,
>
> We thank the reviewer for their observations regarding the practicality of the Quorus framework for enabling practical QFL on near-term processors, as well as the reviewer’s affirmation on the scalability of our technique. Your constructive feedback was very helpful in providing us with clarifications to strengthen our paper, and below, we discuss them as well as our revisions to the paper.
>
> **Partial Aggregation and Seldom Updating of Layers**: We thank the reviewer for their concern regarding the methodology behind our aggregation scheme. We would like to clarify that aggregation of parameters are done based on how many clients have a model with that depth (we formalize this in Algorithm 1 in the paper). Because of this, all layers are aggregated at each round, and there is no fairness gap across clients as each layer is aggregated each round. We hope that this clarifies the reviewers’ concern.
>
> **Differentiation from “Layerwise Parameter Sharing” and “Selective Aggregation”**: We thank the reviewer for their concern about how our technique differs from existing aggregation schemes for layerwise parameter sharing and selective aggregation. We would like to highlight that the primary differentiator between our technique and existing techniques is the demonstration of layer-heterogenous federated learning on quantum computing platforms today, which no prior work has explored. In particular, our development of novel techniques to compute layerwise loss tailored for quantum machine learning models (discussed in Section 4) helps enable practical QFL, which we evaluate on real quantum hardware (discussed in Section 6). We hope that this clarifies the reviewer’s concern.
>
> **Comparison with Communication-Efficient Federated Learning**: We thank the reviewer for referencing communication-efficient classical federated learning for us to compare to. We believe that all of the referenced classical work is tailored for classical machine learning models and thus cannot be directly applied in the quantum federated learning context; this is why these techniques are not directly comparable to Quorus. Below, we outline reasons why, for each of the referenced work:
>
> - FedDrop [1]: This technique requires the ability to drop neurons from some global model, that clients can train. In quantum machine learning, neurons are individual elements of the quantum statevector. To remove individual elements of the quantum statevector, it is highly nontrivial. Below, we discuss two main approaches one would attempt to do this:
>   - Remove qubits from the system: It appears reasonable to remove qubits from the system to remove statevectors from the system. However, due to quantum entanglement, for an arbitrarily entangled system, it is impossible to “factor out” one qubit from your statevector (or the the statevector would not be entangled with that qubit).
>   - Expensive Multi-Qubit Controls: For the above reason, one cannot arbitrarily “drop out” qubits from a quantum ML model without changing other neurons’ behavior. Because of this, to do a “dropout”, it is generally necessary to do a multi-qubit control gate to “omit” a certain neuron from computation. Doing this multi-qubit control gate is very expensive on quantum hardware, and would require clients to have more quantum resources, not less.
>   - In summary, because “dropping out” neurons are not native operations for arbitrary quantum models, this classical technique is not well-motivated in the quantum ML setting, and thus, we do not compare against it.
> - Sparse Ternary Compression [2]: This technique employs sparsification of client updates to enable compression of data sent to and from the server. When considering this technique in the context of quantum federated learning, the fundamental difference in bottlenecks between classical and quantum federated learning arises. In quantum federated learning, the “weight matrix” is implicitly represented through the parameterized unitary matrix; we obtain a significantly smaller representation of the “weight matrix” through the parameters that we tune the unitary matrix with (due to principles in quantum computing). As the number of parameters in our quantum model are very few compared to the size of the weight matrix (for the experiments in our paper, the “weight matrix” is of size $1024^2$, whereas we have $60$ to $180$ parameters for our client models; see Table 6 for details on the experimental setup in the paper), the amount of classical information sent over the classical network is not really the bottleneck because our quantum models have so few parameters. Thus, sparse ternary compression is not well-motivated for our QFL setup, and so we do not compare against it.

---

> > ### Author Response · Authors · 2025-11-21
> >
> > - Selective-FedAvg [3]: We could not find the exact paper referencing this technique. We believe that the reviewer is referring to [3], as their technique is called “S-FedAvg”. This paper solves a different problem than ours: they do not do solve the problem where each client has models of heterogenous depths, and so we do not compare to this technique.
> > - FedPer [4]: This paper solves a different problem than what we are proposing. The problem here allows for clients to have personalized layers local to themselves, which is different from our problem where we want clients to train together with all of their layers. This allows for later layers to be aggregated with parameters with other clients’, which both allows for later layers to receive updates from more clients’ data and is a different setup in terms of the amount of parameters the clients are willing to expose. Thus, we do not compare to this technique.
> > - FedRep [5]: This paper has a similar problem formulation to FedPer, in that later layers are not aggregated. As discussed for FedPer, this choice disables clients from being able to aggregate parameters in later layers to receive updates from more data, and is a different setup from the amount of parameters each client exposes. Thus, this technique is not directly comparable with Quorus.
> > - FedPAQ [6]: Similar to our argument for FedDrop, because FedPAQ is fundamentally tailored for classical machine learning models, it optimizes for reducing the communication complexity of parameters sent to and from the server to clients. Because, in our setup, our parameters are very few due to the implicit representation through our quantum machine learning model, communication of classical parameters is not the primary bottleneck, and so this technique is not directly comparable to Quorus.
> >
> > We have attempted to provide a detailed justification for why each of the referenced techniques is not directly comparable to Quorus, because of their lack of quantum considerations. We hope this addresses the reviewer’s concern.
> >
> > **Hyperparameters**: We thank the reviewer for the concern regarding the sensitivity of our model to hyperparameters related to client selection and aggregation. We would like to clarify that our technique (the algorithm can be found in Algorithm 1) requires no hyperparameters for client selection, as all clients are participating, and no hyperparameters for aggregation, as aggregation is done based on the number of clients that contain a model with that particular depth.
> >
> > **Special Case of Layer-Dropout or Structured Sparsification**:  We thank the reviewer for their question if our technique is a special case of layer-dropout or structured sparsification. We would like to clarify that clients train using all of their parameters and gradient updates are on all parameters; and thus we do not skip layers during training (layer dropout) nor make our model sparse (structured-sparsification). Thus, our technique is not a special case of the proposed techniques; our technique deserves to be treated as a new paradigm because of the quantum-specific considerations we propose as discussed in Section 4 of the paper, with circular parameter aggregation and novel quantum hardware considerations for implementing layerwise loss.
> >
> > We thank the reviewer for their detailed feedback and hope our comments and revisions address your concerns. Please let us know if they do.
> >
> > Sincerely,
> > The Authors
> >
> > References:
> > [1] Wen, Dingzhu, Ki-Jun Jeon, and Kaibin Huang. "Federated dropout—A simple approach for enabling federated learning on resource constrained devices." IEEE wireless communications letters 11.5 (2022): 923-927.
> > [2] Sattler, Felix, et al. "Robust and communication-efficient federated learning from non-iid data." IEEE transactions on neural networks and learning systems 31.9 (2019): 3400-3413.
> > [3] Nagalapatti, Lokesh, and Ramasuri Narayanam. "Game of gradients: Mitigating irrelevant clients in federated learning." Proceedings of the AAAI Conference on Artificial Intelligence. Vol. 35. No. 10. 2021.
> > [4] Arivazhagan, Manoj Ghuhan, et al. "Federated learning with personalization layers." arXiv preprint arXiv:1912.00818 (2019).
> > [5] Collins, Liam, et al. "Exploiting shared representations for personalized federated learning." International conference on machine learning. PMLR, 2021.
> > [6] Reisizadeh, Amirhossein, et al. "Fedpaq: A communication-efficient federated learning method with periodic averaging and quantization." International conference on artificial intelligence and statistics. PMLR, 2020.

---

### Official Review · Reviewer_6SVZ · 2025-11-01

**Soundness:** 3
**Presentation:** 3
**Contribution:** 3
**Rating:** 6
**Confidence:** 3

**Summary:**

This paper proposes a quantum federated learning algorithm that can account for errors in quantum circuits. The authors use extensive simulations to show how this approach can improve test accuracy.

**Strengths:**

+ The idea of using varying-depth quantum circuits is both interesting and timely.
+ The authors provide extensive experiments to validate their approach.
+ The solution can reduce barren plateau effects.
+ There are clear relevance to emerging quantum platforms.

**Weaknesses:**

- The gains from the experiments seem very limited.
- The scalability of this solution is not studied.
- The results seem limited to basic classification tasks.

**Questions:**

- How does your algorithm scale with the number of layers and number of agents? It would be useful to provide scalability experiments.
- Can you handle heterogeneous datasets? Can you run experiments with such settings?
- How do you justify such a complex design for a very small (around 10%) gain?
- What type of quantum hardware is needed for this solution to work?
- Can you study the effect of error correction or mitigation?

---

> ### Author Response · Authors · 2025-11-21
>
> Dear Reviewer 6SVZ,
>
> We are deeply grateful for your detailed feedback on our work. We appreciate your recognition of the timeliness and relevance of our work to current-day quantum platforms, as well as our use of extensive experiments in validating our approach. We also appreciate your recognition of our solution’s ability to reduce barren plateau effects. Your constructive criticism was very helpful, and we hope that our response has addressed your main concerns. Below, we summarize our responses as well as the changes we have made in the paper based on your feedback (highlighted in blue in the paper).
>
> **Limited Gains from Experiments/Complexity of Solution**: We thank the reviewer for their concern that the gains from experiments appear limited for our solution. Looking at existing literature in federated learning techniques, the improvements for our solution are of similar magnitude to the improvements in other papers. In light of this, we believe that the gains from our solution are actually quite significant.
>
> Looking at classical federated learning works cited as Related Work in Section 3, DepthFL [1] (ICLR 2023) shows a 0.2% improvement for MNIST, ScaleFL [2] (CVPR 2023) shows an improvement of 1-3% for HeteroFL, and ReeFL [3] (ICML 2024) shows an improvement of 10-15% compared to SOTA. Our average improvement of 12.4% over Q-HeteroFL is consistent with what is considered significant gains in existing literature.
>
> The reviewer also asked if we could “justify such a complex design”. We would like to seek clarification from the reviewer on where our design is complex. To that end, we have added a detailed clarification of the costs of using Quorus in Appendix I. We believe that our design simply requires clients to be able to compute layerwise losses and change the quantum model that they run. Aside from these, there are no additional costs of using Quorus. We have designed our technique so that there is effectively no quantum resource overhead for a client to use it; rather, we have tailored Quorus for various hardware constraints (discussed in detail in Appendix I).
>
> **How Quorus Scales**: We thank the reviewer for asking about how Quorus scales with the number of layers and number of clients. With respect to how Quorus scales with the number of layers, we have aimed to address this in the current version of the paper in Table 9, where we compare three different experiment types for Quorus: Quorus with 1-5 Layers, Quorus with 2-6 Layers, and Quorus with 2-10 layers. In Appendix D.2, we have discussed how, based on the experimental results, Quorus with fewer layers (1-5 layers) performs worse, likely due to one layer being insufficient for the machine learning task (with significant accuracy degradation compared to when clients have at least two layers, based on our empirical analysis). To further supplement that claim, we have added a theoretical analysis of how our one-layer ansatz behaves with respect to the input in Appendix H. We find that, for the single layer circuit, the loss function is not a function of half of the parameters in the circuit. However, with two layers, due to increased entanglement between qubits, we empirically observe higher testing accuracy. With respect to more layers, we see empirically in Table 9 that Quorus with 2-10 layers sometimes has worse performance compared to Quorus with just 6 layers, suggesting a marginal gain (or even accuracy degradation) in testing accuracy with additional layers. This phenomenon has also been explored in other papers [4, 5], highlighting how increasing the number of layers may not always help. Because of this, we performed Quorus with just 2-6 layers in our evaluation.
>
> Regarding running Quorus with more agents, we argue that, in contrast to classical computing, the number of quantum computers that exist today are relatively few [6]. Thus, we believe that running our technique with many clients is not the use-case we would expect for practical federated learning in the near-term. Thus, we run Quorus with just 5 clients in our evaluation, and highlight that Quorus’s framework is flexible enough to support additional clients.
>
> **Basic Classification Tasks**: We thank the reviewer for their concern that the results appear limited to basic classification tasks. To expand the scope of our evaluation, we have revised the paper to provide two sets of additional results: multiclass classification (4-class) and CIFAR-10 binary classification, in Tables 4 and 5 of the paper, respectively, and analyzed in Section 7. We see that Quorus exhibits an average of 24.0% improvement over QHeteroFL for 4-class classification and an average of 6.7% improvement over CIFAR-10 in more challenging binary classification tasks. In doing so, we verify the robustness of Quorus for challenging classification tasks.

---

> > ### Author Response · Authors · 2025-11-21
> >
> > **Heterogenous Datasets**: We thank the reviewer for the suggestion to provide experiments with heterogeneous datasets; specifically with non-IID data. Evaluating our technique on the same MNIST and Fashion-MNIST binary classification tasks presented in the paper, we split data according to a Dirichlet(0.5) distribution so that data is non-IID across clients. We show our results in Table 10 and discuss them in Appendix G. Comparing Quorus to the Q-HeteroFL baseline, we see that Quorus has an average 22.5% improvement in testing accuracy over the baseline, highlighting its robustness even to non-IID data for real-world federated learning setups.
> >
> > **Quantum Hardware Needed**: We thank the reviewer for their concern regarding the quantum hardware needed to run our technique. Because our V-Shaped ansatz utilizes only nearest-neighbor connectivity, our technique can be run on any quantum hardware provider supporting digital quantum computation, with connectivity constraints, and the ability to do measurement. These requirements are supported by major quantum clients today, such as IBM, Google, QuEra, IonQ, and Quantinuum, to name a few. The base quantum hardware constraints required to run Quorus - Layerwise are minimal. Regarding a detailed discussion of the quantum hardware requirements needed to run the variants of Quorus, we have discussed this in the main paper, but we have added a discussion on this in Appendix I.2 to provide more detailed clarification.
> >
> > **Effect of Error Correction/Mitigation**: We thank the reviewer for their concern about the impacts of error correction and mitigation on our technique. However, we would like to argue that error correction and mitigation are largely orthogonal to the practicality of our technique, and these tools only serve to increase the usability of our technique.
> >
> > We first would like to highlight that current-day hardware is not large enough to deploy error correction at scale, motivating clients to run circuits of depths only to which they can run at high fidelity (with errors being a primary limiting factor). Even with error correction, clients still have motivation to run different depth circuits. There are two primary reasons for this. One is due to the cost of quantum computation. Looking at current-day quantum hardware providers, many hardware providers bill based on the amount of time the quantum computer is used. Thus, even if a client had an error-corrected quantum computer, if it is very expensive to use (quantum computers are; with some quantum hardware providers charging $7000 per hour of usage [8]), then due to a client’s budget, they are incentivized to run circuits of shallower depth that use up less compute time on the quantum computer. In addition, asymptotically deep quantum circuits are known to have barren plateaus in the general case [7], and, as discussed above and in the main text (empirically verified in Table 9), running deeper circuits does not always yield to higher accuracy. For those reasons, having the ability to run circuits of arbitrary depth with error correction does not mean that clients will always be incentivized to run circuits of high depth, motivating the use of our technique.
> >
> > With respect to error mitigation, error mitigation will only serve to help a client run deeper circuits if they desired, but for the same reasons above, the problem that different clients may want to run different depths still exists, which is precisely the motivation for our solution, Quorus.
> >
> > We are sincerely grateful for your feedback, and hope these revisions address your concerns. Please let us know if they do.
> >
> > Best,
> > The Authors
> >
> > References:
> > [1] Kim, Minjae, et al. "Depthfl: Depthwise federated learning for heterogeneous clients." The Eleventh International Conference on Learning Representations. 2022.
> > [2] Ilhan, Fatih, Gong Su, and Ling Liu. "Scalefl: Resource-adaptive federated learning with heterogeneous clients." Proceedings of the IEEE/CVF Conference on Computer Vision and Pattern Recognition. 2023.
> > [3] Lee, Royson, et al. "Recurrent early exits for federated learning with heterogeneous clients." arXiv preprint arXiv:2405.14791 (2024).
> > [4] Sim, Sukin, Peter D. Johnson, and Alán Aspuru‐Guzik. "Expressibility and entangling capability of parameterized quantum circuits for hybrid quantum‐classical algorithms." Advanced Quantum Technologies 2.12 (2019): 1900070.
> > [5] Nguyen, Tuyen, et al. "An evaluation of hardware-efficient quantum neural networks for image data classification." Electronics 11.3 (2022): 437.
> > [6] Tao, Runzhou, et al. "Quantum virtual machines." 19th USENIX Symposium on Operating Systems Design and Implementation (OSDI 25). 2025.
> > [7] Cerezo, Marco, et al. "Cost function dependent barren plateaus in shallow parametrized quantum circuits." Nature communications 12.1 (2021): 1791.
> > [8] https://aws.amazon.com/braket/pricing/

---

### Author Response · Authors · 2025-11-12

Dear Reviewers,

Thank you very much for your thorough feedback and suggestions. We are in the process of reviewing your comments and will provide a response for each of your concerns shortly.

Best,
The Authors

---

### Author Response · Authors · 2025-11-22

Dear Reviewers,

We would like to make a brief note that, based on reviewer comments to include heterogenous data, we have included additional results in Table 10 for Non-IID data for the Fashion-MNIST dataset, with evaluation for Quorus - Funnel (for consistency with the other tables in the paper). We have also clarified and added more detail on how the Non-IID data was sampled in Appendix G. Finally, we have fixed minor typos in Appendix I.2: the number of qubits measured for $C$ class classification is not $log_2(C)$; it is $\lceil log_2(C) \rceil$ (to obtain an integer value for measured qubit count). We have updated our replies to your comments accordingly.

We thank you again for your detailed feedback, and we look forward to your replies.

Best regards,
The Authors

---

### Author Response · Authors · 2025-11-29
**Summary for the Area Chair: Part 1/2**

We thank the reviewers for their exceptionally careful and constructive feedback. Their comments were invaluable and directly shaped substantive improvements to our paper. The revision process materially strengthened our experimental scope, theoretical clarity, algorithmic precision, and practical positioning. As a result, the manuscript is now significantly clearer, more rigorous, and substantially more compelling than the original version submitted.

Given the post-discussion reassignment process, we provide this consolidated summary to ensure that the full scope of our revisions is evaluated holistically. Across all reviews, we directly addressed every concern through new experiments, algorithmic clarifications, theoretical analysis, and expanded discussions.

**Expanded Evaluation Scope and Task Diversity:**
To address concerns about limited evaluation, we added multiclass benchmarks and harder datasets. In **Section 7 and Tables 4–5**, we introduced CIFAR-10 binary tasks showing an average **6.7% accuracy improvement** over Q-HeteroFL and four-class MNIST/Fashion-MNIST tasks demonstrating a **24.0% improvement**. These additions directly resolve concerns that prior results were restricted to simple binary settings.

**Heterogeneous and Non-IID Validation:**
Responding to questions on the realism of client data distributions, we added **non-IID federated experiments using Dirichlet$(0.5)$ splits** in **Table 10 and Appendix G**. Quorus achieves **22.5% higher accuracy** and **11.0% lower standard deviation in accuracy** compared to Q-HeteroFL, demonstrating robustness and stabilization under client heterogeneity.

**Scalability with Circuit Depth and Client Count:**
Depth scalability concerns were addressed with discussion on experiments across regimes (1–5 layers, 2–6 layers, 2–10 layers) in Table 9 and Appendix D.2. Results show performance saturation at moderate depths consistent with expressivity and barren-plateau theory. We formalized this behavior in **Appendix H** with analysis showing that single-layer circuits underutilize model capacity, suggesting that multiple layers are required for additional expressivity. We further clarified that current hardware constraints realistically limit the sizes of active QFL clients, while the Quorus protocol itself remains fully general, based on our results in Section 6.

**Algorithmic Fairness and Aggregation Guarantees:**
Concerns regarding biased aggregation were resolved in Algorithm 1 by explicitly specifying that **every layer is aggregated in every round**, weighted by the number of clients capable of supporting each depth. No layer is skipped, and no client is excluded. This ensures fairness and eliminates any ambiguity regarding selective or partial updates.

**Clear Differentiation from Classical Heterogeneous FL:**
We provided discussion to clarify why classical approaches such as FedDrop, FedPer, FedRep, S-FedAvg, and FedPAQ do not translate to the quantum setting. The constraints imposed by entanglement, measurement collapse, and low PQC parameter dimensionality make the use of drop or compression strategies employed in classical FL prohibitively expensive or unnecessary. Quorus introduces quantum-native advances including **layerwise loss construction, circular depth-aware aggregation, and circuit-level hardware compatibility**.

**KL Coupling and Expressivity Clarification:**
We addressed concerns that KL regularization might restrict the use of deeper models. In Section 6 and Table 2, we empirically demonstrate that KL coupling **increases gradient magnitude and test accuracy** for deeper clients without sacrificing expressive capacity. The coupling stabilizes training where data contributions thin at high depths, rather than acting as a restrictive regularizer.

**Removal of "Single-Shot" Ambiguity:**
All misleading references to "single-shot" inference were removed or clarified. In **Appendix I.2**, we now precisely define prediction efficiency via **obtaining marginal distributions on qubits after each layer as prediction vectors without rerunning circuits per depth**, contrasting this with naïve rerun-based approaches that would scale linearly in shot cost.

**State Handling and Tomography Clarification:**
We clarified in **Appendix I.2.1** that the protocol requires **no quantum state tomography**. Intermediate-depth loss evaluation reruns known prefix circuits rather than reconstructing states, resulting in linear overhead that is less than the exponential overhead associated with quantum state tomography.

---

### Author Response · Authors · 2025-11-29
**Summary for the Area Chair: Part 2/2**

**Reproducibility and Encoding Details:**
To address reproducibility concerns, we fully specified feature preprocessing, PCA reductions, qubit allocations, and data re-upload schedules in **Section 5 and Appendix C**, allowing replication of all experimental conditions. We also note that the source code for the manuscript is attached as supplementary material.

**Hardware Realism and Deployability:**
In **Appendix I.2**, we expanded the discussion demonstrating deployability across real platforms, including IBM, IonQ, Quantinuum, Google, and QuEra. Quorus requires no quantum error correction and is designed for near-term devices constrained by decoherence and connectivity limits. We clarified that even under future QEC, depth heterogeneity remains economically and algorithmically relevant.

**Baseline Validation:**
We added discussion that **Q-HeteroFL is the appropriate state-of-the-art baseline** as the direct quantum analog of HeteroFL, which is a commonly used baseline in classical FL literature. No other existing method addresses heterogeneous-depth QFL with an equivalent formulation. Our QFL technique can be fairly compared with other QFL with similar quantum resources; comparing our technique with classical machine learning models is largely orthogonal to our technique, which provides the following contribution: compared to existing QFL approaches and quantum resources, we can allow for varying depths with higher accuracy, with no additional shot budget.

---

**Concluding Reflection:**
We are sincerely grateful for the depth and rigor of the reviewer feedback. Their critique directly guided substantial experimental expansions, improvements in algorithmic descriptions, and clearer positioning of our contributions relative to both classical and quantum literature. Each concern was integrated into the revision, resulting in a manuscript that is both technically stronger and conceptually clearer. With these targeted additions and clarifications, the revised paper now provides a complete, validated, and practically grounded treatment of depth-heterogeneous quantum federated learning through Quorus.

---

### Meta-Review · Area_Chair_Yyx5 · 2025-12-30

**Summary:**

This work introduced a federated learning framework that trains quantum clients with different circuit depths by applying a layerwise loss. The evaluation results on both benchmarks and multiple IBM QPUs demonstrated that the proposed method outperforms the baselines by 10% or so.

Strength:
1. The idea of heterogeneous QFL is very interesting.
2. The concrete FL procedure with circular angle averaging, parameter slicing by depth, and layerwise loss is impressive.
3. It did extensive experiments on both benchmarks and real IBM QPUs to validate the good performance of the proposed method.


Limitations:
1. One concern is whether the proposed method can be applied to multiple-class classification, such as CIFAR-100. While authors did additional experiments on CIFAR-10, it remains to do experiments on ImageNet and CIFAR-100
2. It would be great to compare the proposed method with more baselines.

In short, the authors already addressed most of the concerns raised by reviewers. Hence, I lean towards accepting this work.

**Reviewer Concerns:**

The authors have addressed the following main concerns.

1. Do more experiments on multi-class classification.

2. Studied how Quorus scales with the number of layers and the number of clients.

3. Do more evaluation on non-IID data.

The authors may need to compare the classical baselines. or at least to clarify why not compare with classical baselines.

**Reviewer Scores:**

Reviewer LKv7 may raise the score since the authors clarified and addressed all the concerns.

Reviewer hx6a may increase the score since the authors did additional experiments on multi-class classification, which is a major concern.

---

### Decision · Program_Chairs · 2026-01-26

Accept (Poster)